# Longitudinal single-cell profiling reveals molecular heterogeneity and tumor-immune evolution in refractory mantle cell lymphoma

Shaojun Zhang [1,10], Vivian Changying Jiang [2,10], Guangchun Han [1], Dapeng Hao[1], Junwei Lian[2], Yang Liu[2], Qingsong Cai[2], Rongjia Zhang [2], Joseph McIntosh[2], Ruiping Wang[1], Minghao Dang[1], Enyu Dai[1], Yuanxin Wang [1], David Santos[3], Maria Badillo[2], Angela Leeming[2], Zhihong Chen[2], Kimberly Hartig[2], John Bigcal[2], Jia Zhou [4], Rashmi Kanagal-Shamanna[5], Chi Young Ok [5], Hun Lee[2], Raphael E. Steiner[2], Jianhua Zhang [1], Xingzhi Song[1], Ranjit Nair[2], Sairah Ahmed[2], Alma Rodriquez[2], Selvi Thirumurthi[6], Preetesh Jain[2], Nicolaus Wagner-Bartak[7], Holly Hill [2], Krystle Nomie[2], Christopher Flowers[2], Andrew Futreal [1], Linghua Wang [1,8,11✉] & Michael Wang [2,9,11✉]

The mechanisms driving therapeutic resistance and poor outcomes of mantle cell lymphoma (MCL) are incompletely understood. We characterize the cellular and molecular heterogeneity within and across patients and delineate the dynamic evolution of tumor and immune cell compartments at single cell resolution in longitudinal specimens from ibrutinib-sensitive patients and non-responders. Temporal activation of multiple cancer hallmark pathways and acquisition of 17q are observed in a refractory MCL. Multi-platform validation is performed at genomic and cellular levels in PDX models and larger patient cohorts. We demonstrate that due to 17q gain, BIRC5/survivin expression is upregulated in resistant MCL tumor cells and targeting BIRC5 results in marked tumor inhibition in preclinical models. In addition, we discover notable differences in the tumor microenvironment including progressive dampening of CD8+ T cells and aberrant cell-to-cell communication networks in refractory MCLs. This study reveals diverse and dynamic tumor and immune programs underlying therapy resistance in MCL.

[1] Department of Genomic Medicine, The University of Texas MD Anderson Cancer Center, Houston, TX, USA. [2] Department of Lymphoma and Myeloma, The University of Texas MD Anderson Cancer Center, Houston, TX, USA. [3] Department of Surgical Oncology, The University of Texas MD Anderson Cancer Center, Houston, TX, USA. [4] Department of Pharmacology and Toxicology, The University of Texas Medical Branch, Galveston, TX, USA. [5] Department of Hematopathology, The University of Texas MD Anderson Cancer Center, Houston, TX, USA. [6] Department of Gastroenterology, Hepathology & Nutrition, The University of Texas MD Anderson Cancer Center, Houston, TX, USA. [7] Department of Abdominal Imaging, The University of Texas MD Anderson Cancer Center, Houston, TX, USA. [8] MD Anderson Cancer Center UTHealth Graduate School of Biomedical Sciences, Houston, TX, USA. [9] Department of Stem Cell Transplantation and Cellular Therapy, The University of Texas MD Anderson Cancer Center, Houston, TX, USA. [10]These authors contributed equally: Shaojun Zhang, Vivian Changying Jiang. [11]These authors jointly supervised this work: Linghua Wang, Michael Wang. ✉email: lwang22@mdanderson.org; miwang@mdanderson.org

Mantle cell lymphoma (MCL) is an aggressive and incurable subtype of non-Hodgkin lymphoma that exhibits marked clinical, pathological, and genetic heterogeneity[1–9]. MCL patients experience disease progression or relapse after almost any therapy. Currently, Bruton's tyrosine kinase (BTK) inhibitor ibrutinib and the BCL-2 inhibitor venetoclax are clinical MCL treatment options that produce high response rates and reasonably durable outcomes in this patient population[10–12]. However, resistance to one or both of these agents frequently develops in MCL, resulting in poor survival outcomes and necessitating a better understanding of the cellular and molecular basis of their resistance[13,14]. Recent work by our group identified genomic and transcriptomic alterations in resistant MCL cells, including metabolic reprogramming toward glutamine-fueled oxidative phosphorylation (OXPHOS) that drives ibrutinib resistance in MCL[15]. Yet, these studies were centered on bulk analysis, and the single-cell landscape and roles of individual cell populations, such as tumor and immune cell subsets and their dynamic interactions, in therapeutic resistance and evolution of MCL, has not been systematically characterized.

The cellular complexity and the dynamic evolutionary character of cancer are key factors contributing to therapeutic failure and disease progression in oncology[16] and are therefore obstacles to improving clinical outcomes. The dynamic interactions and co-evolution of the tumor and tumor microenvironment (TME) in response to hypoxic and therapy-induced stress lead to continuous changes in cellular and molecular properties, culminating in the development of resistance. Therefore, longitudinal tracking of clonal heterogeneity and evolution of the complex tumor ecosystems can advance our understanding of how diverse tumor and immune programs drive therapy resistance and inform novel therapeutic strategies.

In this work, we dissect the dynamic multicellular tumor "ecosystem" using the cutting-edge single-cell transcriptome sequencing (scRNA-seq) technology, coupled with longitudinal sampling and deep molecular profiling to understand refractory MCL at an exceptional resolution.

## Results

**Longitudinal single-cell analysis uncovers dynamic cellular heterogeneity of MCL.** To understand the cellular and molecular mechanisms of refractory MCL, we performed sequential scRNA-seq of 21 specimens (discovery cohort) collected at baseline, during treatment, and/or at disease remission/progression from three ibrutinib-responsive (R) patients (Pt-V, C, and D) and 2 non-responsive (NR) patients (Pt-B and E). In addition, the PBMC samples from two healthy donors (N1 and N2) were included as the normal controls. (Fig. 1a, b and Supplementary Data 1). Patient V has a typical CLL-type MCL in a leukemic phase with splenomegaly and widespread infiltration of bone marrow (BM) but without lymphadenopathy. This patient responded slowly to ibrutinib and achieved partial remission at cycle 12 with gradual shrinkage of the spleen size and parallel decline of lymphocytes in peripheral blood (PB) (Fig. 1b and Supplementary Data 1). Patient C presented with leukemic MCL with lymphadenopathies involving multiple compartments including BM involvement (55%) but without splenomegaly. Patient D presented with typical lymphadenopathy and a moderate BM involvement (19%) but without being in a leukemic phase. Patients C and D achieved complete remission after ibrutinib treatment, at cycle seven and six, respectively. Patient B was diagnosed as stage IV MCL with splenomegaly and multiple-compartment lymphadenopathies, PB (70%), and BM (33%) involvement. Patient B progressed on multiple therapies including ibrutinib and venetoclax. Patient E is an MCL patient

who relapsed from multiple lines of therapies including ibrutinib and venetoclax. This patient presented with splenomegaly, subcutaneous and muscular lesions, and retroperitoneal/pelvic lymphadenopathies, with BM (40%) involvement and strong CCND1 staining in the BM biopsy (Supplementary Fig. 1). Consequently, Patients B and E were defined as refractory MCLs. Altogether, these characteristics demonstrate a high degree of clinicopathological heterogeneity across and within ibrutinib responders and non-responders. Additional clinical characteristics of the patients are provided in Supplementary Data 1.

Of 20,004 sequenced cells, 18,794 (94%) passed quality filtering with an average of ~73,727 reads aligned per cell (Methods). Dimensional reduction analysis (t-SNE, t-distributed stochastic neighbor embedding) and unsupervised clustering were performed to classify the cells based on their transcript expression profiles. The tumor B cells were clustered distinctly from the non-malignant cells of the TME or the normal B cells from healthy donors (Fig. 1c and Supplementary Fig. 2). In addition, the tumor B cells were distantly separated by the patient and therapeutic response (Fig. 1c), followed by sample collection time point (Fig. 2a and Supplementary Fig. 2a), demonstrating a high degree of inter- and intra-tumoral cellular heterogeneity as observed in other cancers. For example, the tumor cells from patients B, C, and V formed separate clusters for each sample collected at different time points during the treatment (Supplementary Fig. 2b). Interestingly, the malignant cells of tumor B4, collected at disease progression from patient B, were separated into two clusters (B4a and B4b) (Fig. 2a, middle), indicating the coexistence of transcriptomically heterogeneous subpopulations within the same tumor. In contrast, the non-malignant cells from MCL patients and normal blood cells were clustered closely by cell type (Fig. 1c and Supplementary Fig. 2), irrespective of tissue sources (Fig. 1c, Supplementary Fig. 2). A longitudinal examination of the cellular composition indicated dynamic changes of MCL ecosystems during treatment, with the relative fractions of tumor B cells increasing in NR (e.g., Pt-B) and decreasing in R (e.g., Pt-V) (Fig. 1d). Consistent with clinical presentation, the fractions of tumor cells in D1 (PB, 10%) and D2 (BM, 25%) before ibrutinib therapy were much lower than those in Patients V (V0–V2, >80%) and C (C1 and C2, >80%) with leukemic MCL and very few MCL cells were detected from the baseline sample (E1) collected from the PB of patient E (Fig. 1d and Supplementary Data 1). Minimal batch effects were detected and we observed no difference in the tumor/immune cell composition pre and post batch effect correction[17] (Fig. 1d and Supplementary Fig. 3).

**Cellular and molecular heterogeneity and evolution of cancer hallmark signaling associates with disease progression and therapeutic resistance.** To identify the transcriptomic features that are common to all MCL cells, we performed differential gene expression analysis between the MCL cells from samples at baseline and the normal B cells from healthy donors and compared the overlap between sets of DEGs across the baseline samples from five patients. We identified 6 downregulated genes and 20 upregulated genes that were ubiquitous to all baseline samples from 5 patients (Supplementary Fig. 4a and Supplementary Data 2). The downregulated genes included PIK3IP1 (a negative regulator of PI3K)[18], and DDIT4 (an inhibitor of mTORC1 signaling)[19]. The upregulated genes included CCND1, STMN1 (also named oncoprotein 18, frequently expressed in high-grade lymphoma)[20], MARCKS (the major protein kinase C substrate that regulates PI3K/AKT signaling)[21,22], FCRLA (a tumor-associated antigen of BCL)[23], FCRL2 (a prognostic marker

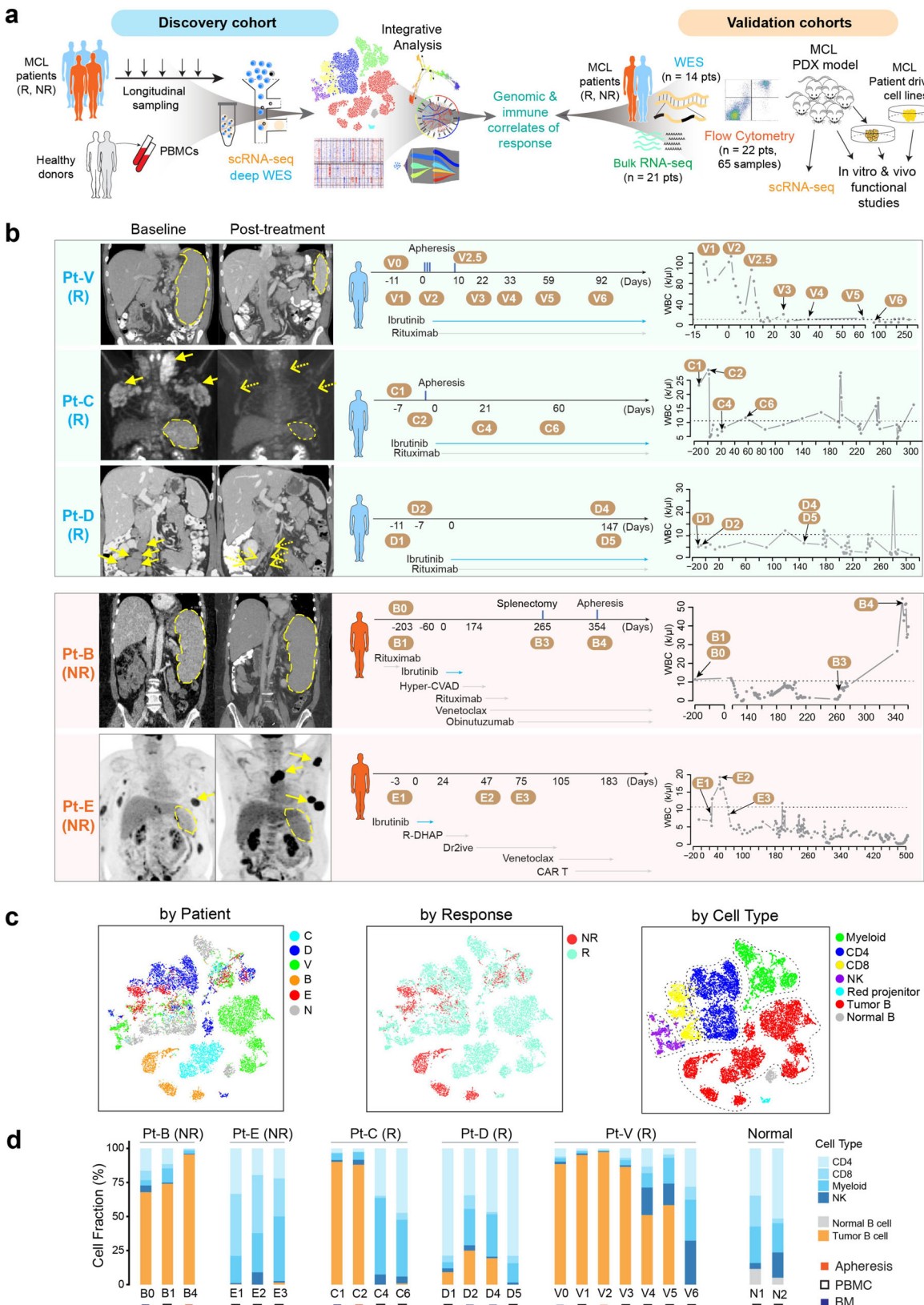

of CLL with a strong correlation with mutated IGHV status)[24], and *VPREB3* (a pre-B-cell receptor (BCR) associated protein and a diagnostic marker for identifying c-MYC translocated lymphomas)[25]. We note that the expression levels of *STMN1 and MARCKS* were significantly elevated in MCL cells from B4 at disease progression (Supplementary Fig. 4b), suggesting a

potential role of *STMN1 and MARCKS* in promoting MCL progression.

Next, we investigated the proliferative heterogeneity of tumor cells via cell-cycle phase-specific signatures[26], based on which a cell cycle stage was computationally assigned to each cell. The complexity of proliferative heterogeneity was revealed in the

**Fig. 1 Longitudinal scRNA-seq of MCL during treatment. a** Schematic view of the experimental design to delineate therapeutic resistance of MCL. The discovery cohort included scRNA-seq and deep whole-exome sequencing (WES) of MCL cells collected longitudinally from three responders (Rs) and two nonresponders (NRs), together with normal PBMCs from two healthy donors. The genomic and immune correlates of response identified from the discovery cohort were then cross-validated by multiple platforms including bulk RNA-seq, WES, and flow cytometry of independent patient cohorts, scRNA-seq of patient-derived PDX models, as well as in vitro and in vivo functional studies using MCL patient-derived cell lines. **b** Coronal or axial images from CT scans pre-ibrutinib treatment (baseline) and post-treatment (disease progression) (left). Spleen sizes were measured and labeled with schematics of treatment and sample collection time points for scRNA-seq (middle); as well as the kinetics of white blood cell (WBC) counts during the course of treatment (right). Specimens were collected at multiple time points before and during the treatment when feasible, including pretreatment, on-treatment, and progression samples. **c** A t-SNE overview of the cells that passed quality control. Each dot of the t-SNE (t-distributed stochastic neighbor embedding) plot represents a single cell. Cells are color-coded by subject (MCL patients B–E, V, and healthy donors N1/N2, cells are merged as N), ibrutinib response status (NR: non-responder; R: responder), and by the cell type. **d** Cell composition dynamics at different time points during sample collection.

---

fraction of proliferating cells across and within patients, particularly the tumors from patient B at disease progression (Fig. 2a). For example, the smaller cell cluster B4b was a highly proliferative tumor with cells at either the G2/M, or S phase and displayed high expression of *MKI67*, *CDK4*, and other proliferative markers such as *PCNA*, *TK1*, and *TYMS* (Supplementary Fig. 5a). However, tumor cells present in the larger cell cluster B4a were relatively quiescent, indicating striking molecular heterogeneity in sustaining proliferative signaling in cancer cells.

To assess whether the dynamic variability and ITH of tumor cells affect the manifestation of cancer hallmarks and the development of therapeutic resistance and progression in refractory MCLs, we evaluated the pathway activity for 50 cancer hallmark gene sets curated by the Molecular Signature Database (MSigDB) and profiled transcriptomic heterogeneity of the cancer hallmarks associated with ibrutinib resistance. Overall, 13 cancer hallmarks were significantly upregulated in the MCL cells from patient B (the non-responder), including MYC, OXPHOS, BCR, mTORC1, cell cycle, and PI3K/AKT/mTOR signaling (Fig. 2b, c and Supplementary Fig. 5b, c). We observed dynamic activation of these signaling pathways, with upregulation over time in patient B and downregulation in responders (patients C, D, V) in response to therapeutic pressure (Fig. 2b and Supplementary Fig. 5d).

**Cellular and molecular mechanisms of ibrutinib-venetoclax dual resistance in patient B.** The B4-derived subpopulations B4a and B4b were disparate with more strikingly increased activity of cancer hallmark pathways in B4b (Fig. 2c). For example, the smaller cell cluster B4b was a highly proliferative tumor with cells at either the G2/M, or S phase (Fig. 2a, right two panels) and displayed high expression of *MKI67*, *CDK4*, and other proliferative markers such as *PCNA*, *TK1*, and *TYMS* (Supplementary Fig. 5a). However, tumor cells present in the larger cell cluster B4a were relatively quiescent, indicating striking molecular heterogeneity in sustaining proliferative signaling in cancer cells.

Moreover, the noncanonical NF-kB signaling was exclusively activated in B4a, with significantly upregulated *BIRC3* (adjusted p value = $3.48 \times 10^{-11}$), an important upstream regulator for noncanonical NF-kB signaling, *NFKB2* and *RELB* (Fig. 2c), which are downstream transcription factors of NF-kB signaling. Altogether this indicates that the two progressive subpopulations of tumor B4 selectively evolved by utilizing different signaling pathways to promote clonal evolution.

To validate the role of these key signaling pathways in maintaining the very aggressive behavior of tumor B4, we established a PDX mouse model using B4 tumor cells (B4-PDX) which faithfully recapitulated the observed splenomegaly, hepatomegaly, and involvement of BM and PB observed in the NR patient B. We next performed scRNA-seq of the B4-PDX tumors (including circulating PDX cells and PDX disseminated cells

collected from the BM, liver, and spleen) using the same protocol and sequencing platform (Fig. 2d). Integrative analysis of scRNA-seq data from B4 and B4-PDX suggested that tumor cells from the B4-PDX model represented the spectrum of cellular and molecular heterogeneity that was similar to the parental B4 tumor cell populations (Fig. 2e, f, Supplementary Fig. 6). The B4-PDX tumor cells formed two distinct cell clusters based on their expression profiles (Fig. 2d, f). The minor cluster transcriptomically resembles the subpopulation B4b, which is comprised of proliferating MCL cells indicated by high expression of cell cycle-related signature genes such as *MKI67* and *CDK4* (Fig. 2f) and increased levels of activity of OXPHOS, mTORC1, and MYC signaling pathways (Fig. 2e and Supplementary Fig. 5e). In contrast, the transcription profile of the major cluster was suggestive of a relatively quiescent cell phenotype, very similar to that of B4a, with no or low expression of cell proliferation genes, but high expression of *BIRC3*, *NFKB2*, and *RELB* (Fig. 2f) indicating activation of non-canonical NF-kB signaling in this cell population. Our combined data analysis of the patient and PDX tumor cells demonstrate a single-cell landscape of the cellular and molecular heterogeneity in NR patient B and confirmed the essential roles of these cancer hallmarks in MCL tumor formation, therapeutic resistance, and dissemination.

**Temporal clinical and integrated genomic profiling reveals distinct molecular features between ibrutinib-induced lymphocytosis and clonal tumor evolution.** Ibrutinib has been shown in CLL and MCL to induce malignant cell redistribution from the tissue compartment (spleen and lymph node) into the PB during the initial weeks of therapy[12,27,28], a process also called ibrutinib-induced lymphocytosis. Prolonged lymphocytosis is common after ibrutinib treatment and has been associated with favorable prognostic features[29] and considered a surrogate marker for treatment sensitivity[12,28]. Clinically, ibrutinib-induced lymphocytosis is characterized by a rise in the absolute lymphocyte count (ALC) in the PB rapidly following treatment initiation. In this study, increased ALCs were observed shortly after ibrutinib treatment in all of the ibrutinib responders (C, D, V) but not in the NR B patient (Fig. 3a, b, representative examples). We performed a temporal profiling of the clinical, genomic, and transcriptomic features as well as cellular heterogeneity to comprehensively characterize ibrutinib-induced lymphocytosis in MCL patients.

In patient V (R), gradual splenomegaly shrinkage and multiple ALC peaks were observed following ibrutinib treatment at days 2 (V2), 10 (V2.5), and 22 (V3) after treatment (Fig. 3c) and documented by the positron emission tomography/computed tomography (PET/CT) imaging (Fig. 3a). In support of our clinical observation, deep WES analysis of somatic mutations in longitudinal samples from patient V identified a cluster of subclonal somatic mutations in V1 that disappeared in V2, and a new cluster of subclonal mutations in V2 that were not detected

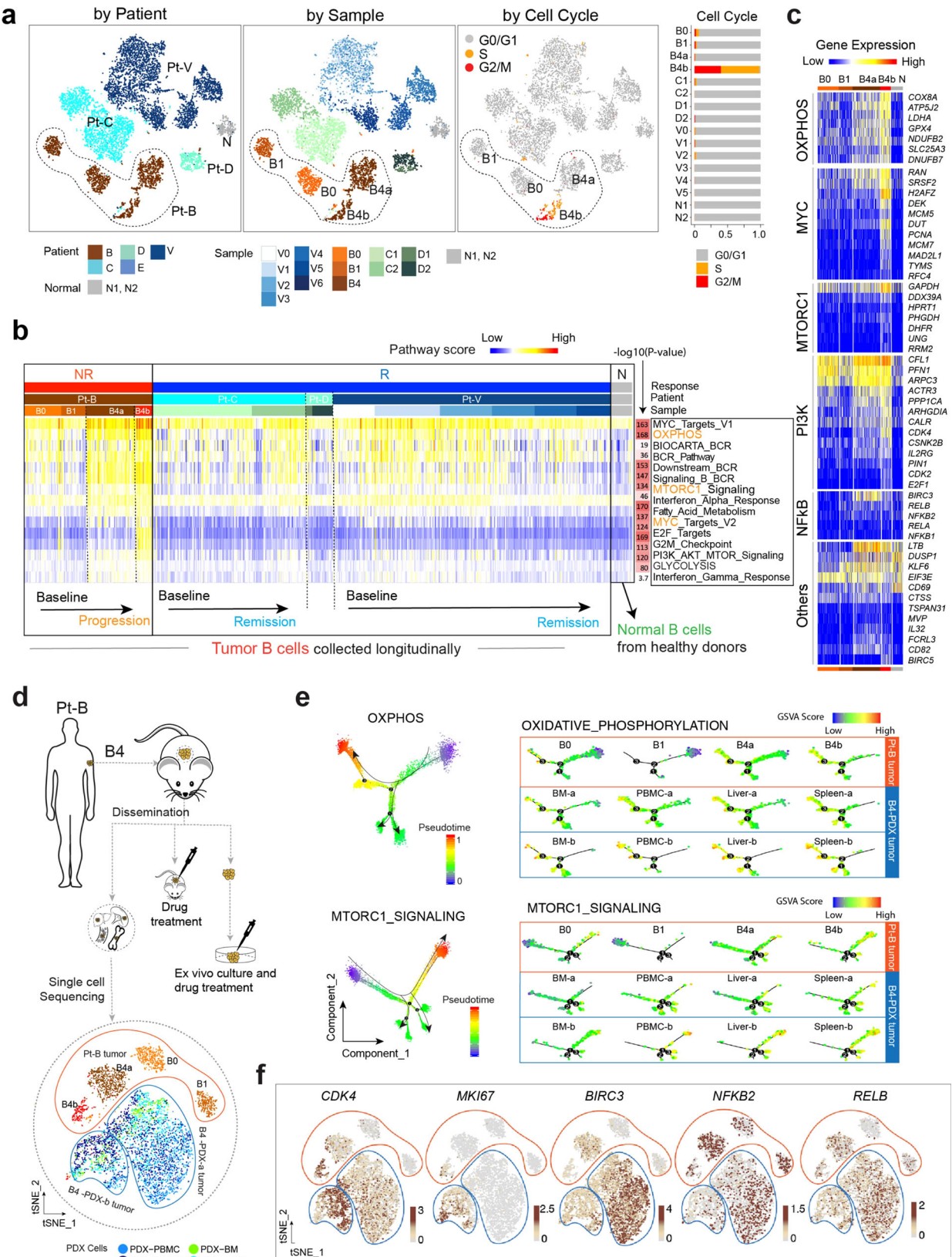

in V1 (Fig. 3e, Supplementary Data 3). Similarly, the cluster of subclonal mutations in V2 disappeared in V2.5 and a new cluster of subclonal mutations emerged, unique to V2.5 (not present in V1 or V2) (Fig. 3e and Supplementary Fig. 7a), indicating sequential redistribution of distinct tumor subclonal populations from compartments, likely spleen, into PB and effective

elimination of these populations within the first 10 days of ibrutinib treatment.

In contrast to the V patient, ibrutinib failed to induce lymphocytosis in the NR Patient B, as indicated by the flat ALC curve (Fig. 3d) with no change in the spleen size (Fig. 3b). In accordance with this notion, we observed the acquisition and

**Fig. 2 The transcriptomic heterogeneity and evolution of cancer hallmarks is associated with therapeutic resistance. a** Color-coded t-SNE plots of the malignant B cells. Color-coded cell representation by subject (Responders: C, D, V; non-responders: B; and healthy donors: N1, N2), sample collection time, and by cell cycle stage. **b** Transcriptomic heterogeneity and evolution of cancer hallmarks associated with ibrutinib resistance. From the Molecular Signature Database (MSigDB), 50 hallmark cancer gene sets were downloaded, and a pathway activity score was calculated for each single cell. The top 13 cancer hallmark pathways upregulated in the progressive sample are shown. **c** Heatmap representation of differentially expressed genes (representative ones) from five selected pathways across cell sub-populations B0, B1, B4a, and B4b from patient B compared to normal samples (N). **d** Schematic view of the establishment of B4-derived PDX model and experimental strategy of sample collection for scRNA-seq analysis. **e** Developmental trajectories representation of malignant cell populations from patient B (B0, B1, B4a, and B4b) and B4-derived PDX tumors along inferred pseudotime by Monocle2. Each point corresponds to a single cell; all points are color-coded according to the inferred pseudotime. Monocle 2 was run with default parameters on the hallmark gene sets (OXPHOS and mTORC1 signaling) downloaded from MSigDB. **f** tSNE Plots featuring *CKD4, MKI67, BIRC3, NFKB2* and *RELB* genes expression in cells from patient B tumor and B4-derived PDX tumors.

clonal expansion of a somatic 17q gain in progressive tumor B3 (collected at splenectomy), which was maintained in B4, and subclonal expansion of a clone carrying *RAD50* somatic mutation from B3 to B4 (Fig. 3f, and Supplementary Fig. 7a, b). Overall, these observations indicate the existence of a therapy-induced clonal expansion and an increase in the degree of genomic ITH during treatment, which is genomically distinct from ibrutinib-induced lymphocytosis.

We further characterized the single-cell transcriptomic features for the two representative cases (Fig. 3g–j). Single-cell clustering analysis with Seurat revealed that the cells of V2 were transcriptomically similar to V1, but cells of V3 were very distinct from those of V0–V2 (Fig. 3g, left). We then applied SC3[30], a different approach for unsupervised single-cell consensus clustering analysis and o1bserved very similar results (Fig. 3g, right, Supplementary Data 4). Cells of V3 were clearly separated from cells of V0–V2, V5, which were clustered together with similar features. Interestingly, the V0/1/2 tumor cells were likely eliminated by ibrutinib at the time point of V3 collection as no cells were detected in the sample V3 that exhibited similar expression features with cells of V0/1/2. Similarly, the vast majority of the V3 tumor cells might have been cleared from PB at the time point of V4 collection as only a small fraction of cells remained (the V4 cells that clustered together with cells of V3) (Fig. 3g, left). The single-cell trajectory analysis clearly demonstrated V3/V4 as different branches that contribute to a tumor path development in response to ibrutinib treatment (Fig. 3i), indicating that new subpopulations of tumor cells redistributed from the tissues into the PB, likely from the spleen.

Distinct from patient V, our analysis suggested the notion that tumor cells in B0 (BM) and B1 (PB) at baseline evolved into two individual subpopulations B4a and B4b at disease progression (Fig. 3h), supported by both Seurat (Fig. 3h, left) and SC3 clustering analysis (Fig. 3h, right, Supplementary Data 4). Lastly, both V and B baseline tumors had clonal *CCND1* mutations as detected by the scRNA-seq and confirmed by the bulk WES (Supplementary Fig. 7c). The *CCND1* E277* mutation found in the founder clone in sample B0 and the *CCND1* E36K mutation identified in V0 was maintained through the subsequent sample collections over the course of therapy (Supplementary Fig. 7c).

**Temporal acquisition of 17q gain in NR patient B.** To understand the potential driving source of the observed transcriptomic heterogeneity, we applied a computational approach (inferCNV) to infer large-scale copy number variations (CNVs) based on the scRNA-seq data. Unsupervised clustering demonstrated very distinct CNV profiles across patients with some shared features within patients, demonstrating greater inter-patient than intra-tumoral heterogeneity (Fig. 4a). We noted that tumor cells of B4 were clustered into two primary clusters based on the inferred CNV profiles (Fig. 4b and Supplementary Fig. 8a), which largely correlated with the transcriptome-based classification of

subpopulation B4a and B4b. We performed integrative analysis to link the inferred genomic alterations and transcriptomic phenotypes. The tumor cells in B4 at disease progression showed notable differences in its CNV profiles with multiple acquired copy number gains and losses, especially gains of chromosomal regions 12p and 17q (Fig. 4a), compared to its baseline samples (B0/B1). Intriguingly, the 17q gain was nearly exclusive to B4 tumor cells and not observed in cells from responsive patients, indicating that 17q gain may have contributed to therapeutic resistance and disease progression of B4 tumor. Notably, the 17q gain was also identified in all disseminated B4-PDX tumor samples (spleen, liver, PB, and BM) derived from B4 by scRNA-seq (Fig. 4c). To validate that this finding truly occurred at the genomic DNA level, we performed deep whole-exome sequencing (WES, mean target coverage: 693×) on three samples collected from patient B, baseline B0, B3 (an additional sample collected at splenectomy, 3 months prior to B4; also at disease progression), and B4 samples, and three samples from patient V (V1, V2, and V2.5). We confirmed the presence of acquired 17q gain in progression tumors B3 and B4, but not in the baseline sample B0 (Supplementary Fig. 8b, c). Consistently, we did not observe 17q gain in samples from patient V (Supplementary Fig. 8b–d), suggesting that this chromosomal alteration might have occurred during the evolution of the tumor to a more aggressive state in patient B.

To further validate the possible correlation of 17q gain with therapeutic resistance in refractory MCLs, we inquired our WES datasets on MCL samples. We identified 17q gain in additional refractory tumors and in an intrinsically ibrutinib-resistant MCL cell line Z138 (Fig. 4d). We did not observe the 17q gain in any of the responsive tumors tested, indicating that the presence of 17q gain may be exclusive to refractory MCLs. Together, our data indicated that the 17q gain could be a driver that emerged during the development of therapeutic resistance in MCL.

Furthermore, we examined the downstream molecular consequence of 17q gain and identified a list of 17q genes with significantly higher expression in B4 against all other samples, especially in the cycling cells from the B4B subpopulation (Fig. 4e, Supplementary Fig. 9a, b left; Supplementary Data 5). A few of these upregulated genes at 17q are known to be involved in important oncogenic signaling pathways, such as *SMARCE1, CBX1, BIRC5* (survivin), *MIEN1, TK1, HN1,* and *TRAF4*[31–38]. *BIRC5* is a gene located at 17q and its overexpression has been associated with poor clinical outcome, cell cycle regulation, and therapeutic resistance[39]. *SMARCE1* and *HN1* have been reported to regulate the metastatic potential of cancer cells[31,32,37]. *CBX1* functions as an oncogene and promote tumor progression in liver cancer[36]. Overexpression of *MIEN1* and *TRAF4* has been reported in various cancers and functionally regulates the PI3K/AKT pathway to promote tumorigenesis[40,41], and therefore, this overexpression may contribute to the dysregulated PI3K/AKT signaling pathway observed in refractory MCLs. Thymidine

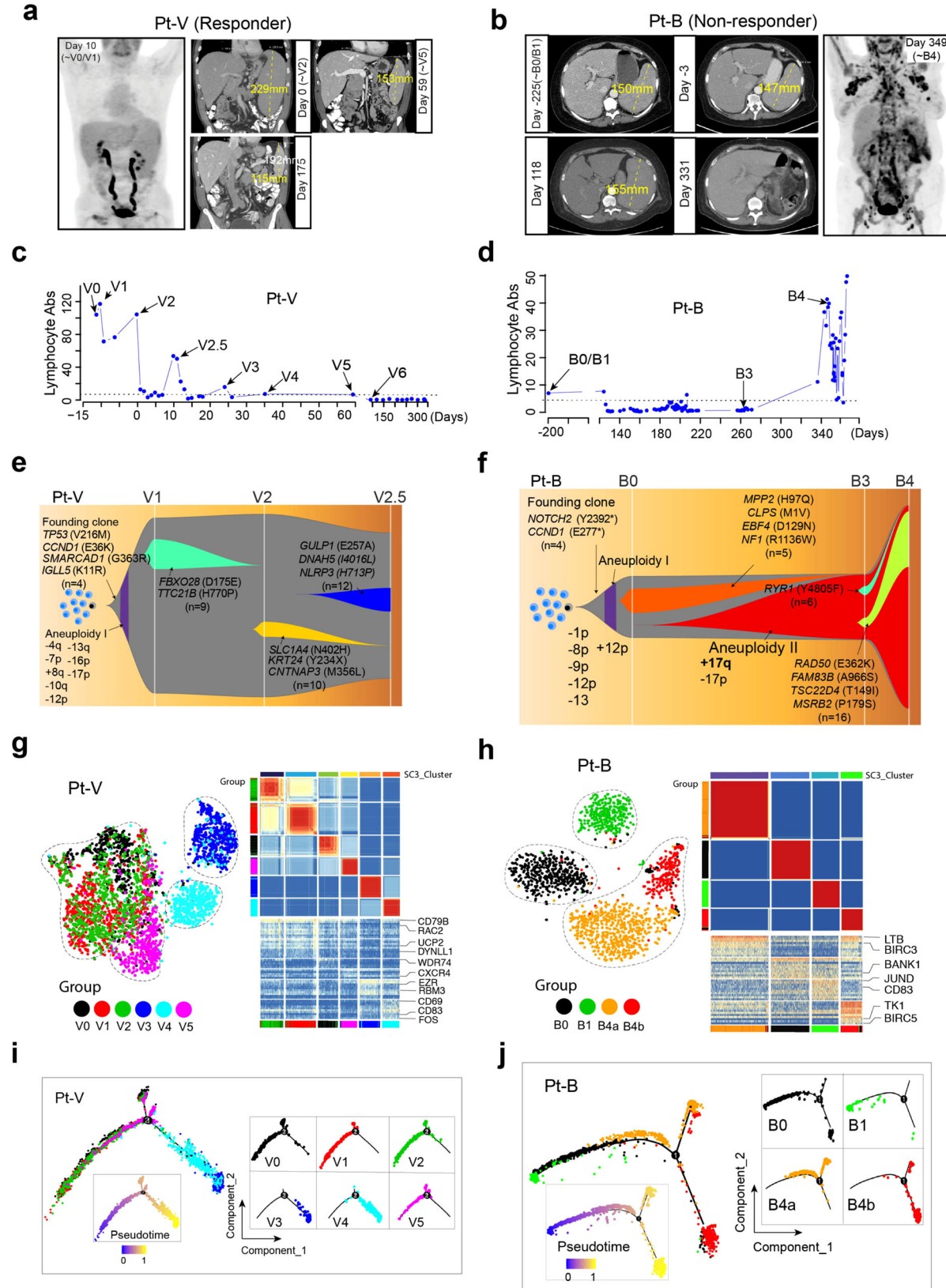

kinase 1, encoded by *TK1*, is a nucleotide enzyme involved in cell proliferation and often considered as an early tumor detection marker[42], may promote cell proliferation of ibrutinib-resistant MCL cells. These results suggest that genomic and transcriptomic ITH are correlated and may cooperatively contribute to disease progression and therapeutic resistance.

**Survivin overexpression at 17q associated with ibrutinib-venetoclax dual resistance in MCL.** Our data (Fig. 4) suggest that the 17q gain may contribute to disease progression and therapeutic resistance in tumor B4. 17q gain was more frequently observed in the B4b than the B4a subpopulation 53% vs. 35%, fisher test $P = 7.04e{-}07$) (Supplementary Fig. 9b, left).

**Fig. 3 Cellular and transcriptomic characterization of ibrutinib-induced lymphocytosis in patient V and clonal evolution in patient B. a, b** The coronal or axial images from PET/CT scan pre- and during ibrutinib treatment. The size of the spleen was measured and labeled. **c, d** The kinetics of lymphocyte absolute count (Abs) measured during the course of treatment. Samples subjected to scRNA-seq are labeled. **e, f** Fish plots showing patterns of clonal evolution of tumors from patients B and V. Clonal evolution was inferred using somatic mutations and DNA copy number alterations identified by deep WES. The representative alterations of each clone are labeled. **g, h** t-SNE plots (left) and SC3 clustering (right) showing the cellular and transcriptomic characterization of the spleen compartment shift of tumor cells during treatment in patient V (left) and therapy induced evolution in patient B (right). **i, j** The developmental trajectories of tumor cells along pseudotime in a two-dimensional state-space inferred by Monocle2.

*BIRC5* (encodes survivin*)*, located at 17q, is highly expressed by the B4b subpopulation (Fig. 5a), especially in the B4b cells at G2/M phase (Supplementary Fig. 9b, middle). The fraction of survivin+ cells was present in 0.3% of the B4a population, 45.6% of the B4b population, and 36.7% of the B4b cells at G2M phase, and survivin expression was strongly associated with 17q gain (Fisher test $P = 2.23e-10$, Supplementary Fig. 9b, right). Our observation is in line with the previous studies showing that survivin inhibits cell apoptosis and promotes cell proliferation at G2/M[39].

To functionally validate the potential role of survivin in maintaining the aggressive behavior of tumor B4, we utilized the established B4-PDX mouse model and performed an integrative analysis of the scRNA-seq data generated from patient B tumors and the B4-derived PDX tumors (Fig. 2d). As described above, we observed a marked increase in survivin expression in the B4b subpopulation in the patient tumor and notably, the high level of survivin expression was maintained in all disseminated tumors of the B4-derived PDX model (Fig. 5a). In support to the observation that survivin expression was enriched in tumor B4b cells at the G2/M phase, single-cell trajectory analysis of MCL cells from patient B tumors (B0, B1, B4) and the B4-derived PDX tumors showed a developmental path towards a significantly increased G2M checkpoint signaling activity in survivin-high tumors (Fig. 5b). Moreover, we examined cell proliferation properties and observed an increased proportion of actively proliferating tumor cells (G2/M or S phase) in B4b and B4-PDX tumors (Fig. 5c), indicating a strong correlation between survivin upregulation and rapid cell division.

We further tested the association between survivin expression and ibrutinib resistance in a second independent MCL patient cohort ($n = 21$) that included 15 ibrutinib-responsive and 6 NR tumors and confirmed higher survivin expression in nonresponders (Fig. 5d). In addition, survivin has been associated with clinical resistance and poor outcome in many cancer types. Indeed, high survivin expression was strongly correlated with significantly shortened survival in MCL patients, revealed by survival analysis in two independent MCL cohorts[43,44] ($n = 71$ and 92, respectively) (Supplementary Fig. 10a). These results indicate an essential role of survivin in driving MCL progression and resistance.

**Targeting survivin overcomes ibrutinib-venetoclax dual resistance in MCL.** Based on our evidence showing that survivin was significantly upregulated in the refractory MCL tumor cells and closely associated with cell proliferation property, we next determined the anti-tumor effects of survivin inhibition in MCL cell lines using the clinically tested survivin inhibitor YM155. (Fig. 5e and Supplementary Fig. 10b). The cell line panel included ibrutinib- and venetoclax-sensitive (Mino and Rec-1), ibrutinib-resistant (Maver and Z138), and venetoclax-resistant (JeKo BTK KD) MCL cell lines. YM155 displayed potent anti-MCL activity ($IC_{50} = 5-30$ nM) in vitro (Fig. 5e) indicated by a significant G1 cell cycle arrest and by induction of cell apoptosis (Supplementary Fig. 10b) across a wide panel of MCL cell lines. We further assessed the anti-tumor effects in PDX models (Fig. 5f–h,

Supplementary Fig. 10c–f). Consistently, YM155 effectively inhibited in vivo subcutaneously implanted tumor growth of the ibrutinib-venetoclax dual resistant B4-PDX model compared to the vehicle-treated control (Fig. 5f, top) and significantly extended mouse survival (Fig. 5f, bottom, Supplementary Data 6) with only one cycle of 28-day continuous infusion at doses as low as 1.0 mg/kg or 3 mg/kg. Moreover, the effects of YM155 on the observed splenomegaly, hepatomegaly, and PB/BM involvement in the B4-PDX mouse model were examined in the ibrutinib-venetoclax dual resistant dissemination PDX model (Fig. 5g, h and Supplementary Fig. 10d–f). YM155 significantly reduced the spleen and liver sizes (Fig. 5g) as well as B2M production in the mouse serum (Supplementary Fig. 10f) in the dissemination model. B2M is a prognostic marker for MCL and serves as an indicator of MCL tumor burden in mouse PDX models[45]. The YM155-treated B4-PDX mice (both animal cohorts) did not show any noticeable toxicities. For example, bodyweight was not significantly different between vehicle control and YM155 treatment group (Supplementary Fig. 10f). These data demonstrate that targeting key cell cycle regulators such as survivin effectively overcomes ibrutinib-venetoclax dual resistance and that the discovery of specific signaling pathways upregulated in individual patients may lead to the development of tailored treatments to overcome therapeutic resistance.

**TME heterogeneity and evolution associated with therapeutic resistance.** The TME acts as a supporting "ecosystem" for tumor growth and progression. To further understand the cellular heterogeneity of TME and the complex interplay between tumor and immune cells, we performed single-cell analysis of the non-malignant immune cells from our discovery cohort. The immune cells from MCL patients clustered separately from the immune cells of healthy donors that were sequenced on the same batch (Fig. 6a and Supplementary Fig. 2a). The clustering results before and after batch effects correction showed no significant difference (Supplementary Fig. 3b, c), indicating possible transcriptomic reprogramming of TME cells. Notably, we observed a significant decrease in the proportion of effector CD8+ T cells in the PB in non-responders post-ibrutinib treatment, in contrast to a dynamic increase in the proportion of effector CD8+ T cells in the samples collected throughout therapy in the responsive patients (Fig. 6b). In support of this observation, the bulk RNA-seq data of our validation cohort (Fig. 1a) showed that CD8A expression was indeed significantly lower (Fig. 6c) in the ibrutinib nonresponders, suggesting that decreased CD8+ T cell-mediated antitumor immunity likely a contributing factor to the observed therapeutic resistance.

To validate this finding in an independent MCL cohort, we performed flow cytometry analysis on 65 samples collected from 22 patients with no overlap with the discovery cohort (Fig. 6d). Consistently, the fraction of CD8+ T cells was significantly lower in samples of the ibrutinib nonresponders, with no significant changes observed in the fraction of total T cells or CD4 T cells between the responders and nonresponders (Fig. 6d), suggesting that CD8 T-cell depletion most likely contributed to therapeutic resistance in refractory MCL. We further examined the

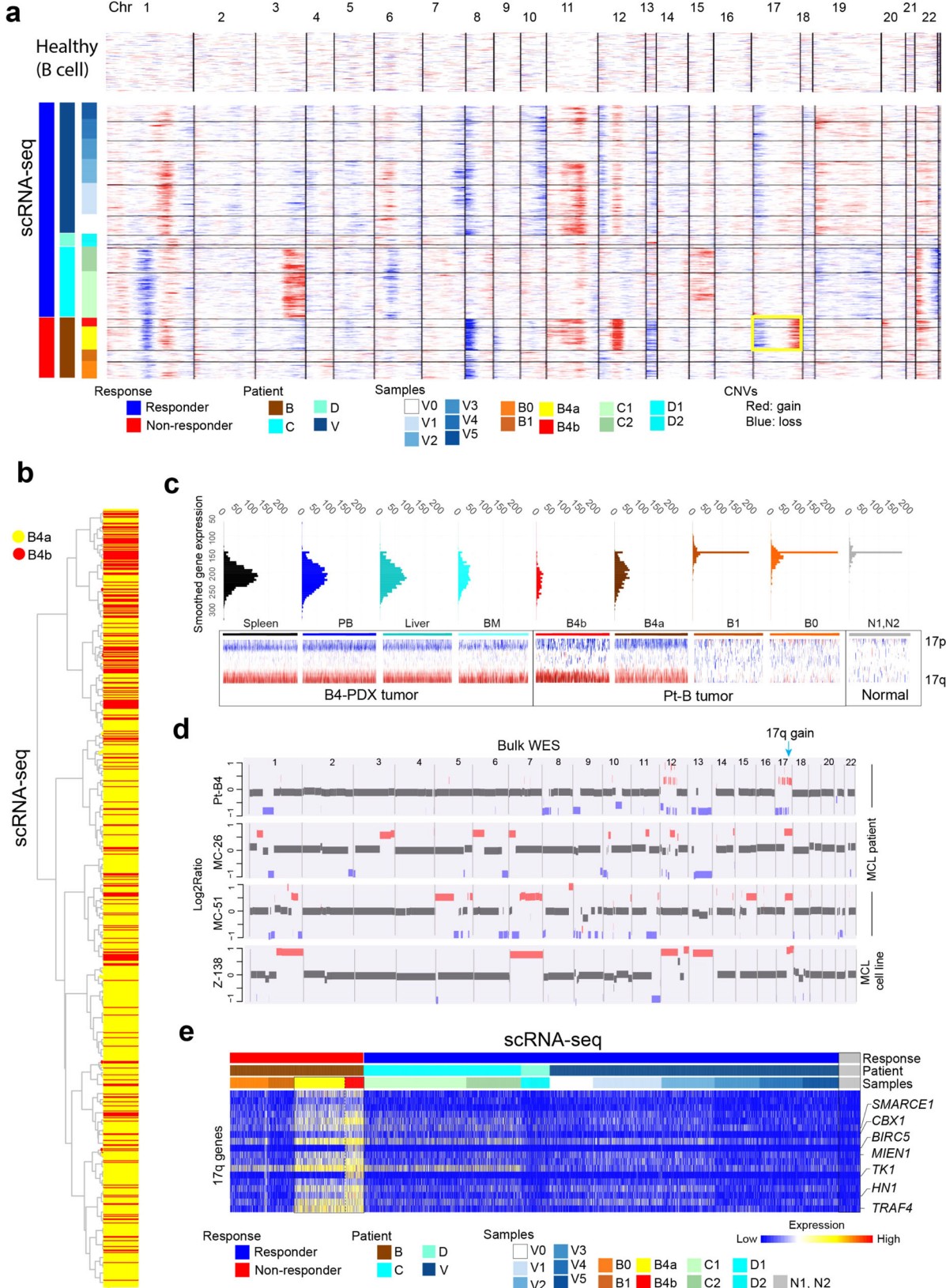

relationship between the activity of oncogenic pathways dysregulated in refractory MCL (Fig. 2b) and the abundance of CD8 T cells, and observed a strong negative correlation between the level of OXPHOS activity in MCL cells and CD8A expression as well as the proportion of CD8+ T cells in the TME (Fig. 6e), indicating that increased OXPHOS activity in MCL cells may

have reprogrammed the TME, but this will require further investigation.

We further analyzed differentially expressed genes (DEGs, NR vs. R) in CD4 and CD8 T-cell subsets at pre- and post-treatment (Supplementary Data 7). Our analysis revealed dramatic changes in the transcriptomic profiles of these cells, particularly in the

**Fig. 4 DNA copy number alterations and heterogeneity is associated with therapeutic resistance. a** Heatmap overview of the inferred copy number alterations (CNAs) in the malignant B cells across 22 chromosomes. Information on patient response status, patient and sample collection time point, were annotated in the left tracks. The yellow rectangle highlights the 17q copy number gain significantly enriched in the progression tumor B4. **b** A dendrogram based on the global CNV profiles showing intra-tumor cellular heterogeneity in B4 tumor cells between two subpopulations B4a to B4b. **c** ScRNA-seq validation of 17q gain in cells from B4-derived PDX tumors. **d** Cross-platform validation of the 17q gain in additional patient cohorts and resistant MCL cell lines using deep whole-exome sequencing (WES). The Log2Ratio plots of 4 representative samples are shown. **e** Expression heatmap showing genes upregulated in the progression tumor B4 and located at 17q.

expression of phenotypic markers and genes associated with cellular functional states (Fig. 7a, b). Interestingly, we observed a significant increase in the gene expression levels of markers related to T regulatory cells (Treg) such as *CXCR4* and *LGALS1* (galectin-1), decreased expression of naïve T-cell-related markers such as *CCR7*, and decreased expression of effector CD8 T-cell markers such as *GZMK*, particularly in the pretreatment samples, suggesting predictive potential for therapeutic response of these markers.

Cell-to-cell communication analysis revealed significant changes (NR vs. R) in a number of signaling networks (Fig. 7c) including the increased inter-cellular interactions between MCL cells and other TME cells including CD4, CD8 T cells and monocytes via galectin-1->*CXCR4/CD69*, and *TGF-β1*->*CXCR4* ligand receptor-based interactions. Galectin-1 functions as a soluble mediator and is employed by tumor cells to evade the immune response and promotes tumor progression[46] through the upregulation of *CXCR4* via NF-kB signaling[47]. Moreover, *CD69* controls T-cell differentiation through its interaction with galectin-1[48], and TGF-β-triggered *CXCL12/CXCR4* signaling has been demonstrated to promote tumor invasion, metastasis, and therapeutic resistance[49]. Flow cytometry confirmed the increased fraction of *CD69*+ T cells both pre- and post-treatment in the non-responders. Moreover, *CXCR4*+ cells were highly represented in the non-responders after treatment in both the CD4 and CD8 T cells (Fig. 7d). In addition, the *PRF1*+ *CD8*+ T-cells (effector T cells) inversely correlated with *CXCR4* expression on CD8 T-cells (Fig.7e), suggesting that *CXCR4* may suppress T-cell function in MCL, but this will require further investigation. Together, our results suggest that complex interactions between MCL cells and the TME may largely influence therapeutic resistance, warranting the development of strategies to promote the anti-lymphoma activity of the TME.

## Discussion

MCL is a rare refractory disease subject to relapse. Novel technologies such as single-cell sequencing is underutilized in MCL to reveal tumor heterogeneity, to track clonal evolution and to discover resistance mechanisms. The *BTK* inhibitor ibrutinib and the *BCL-2* antagonist venetoclax are routinely used in practice for the treatment of relapsed/refractory MCL and are increasingly used in newly diagnosed MCL in clinical trials. However, the development of dual resistance to ibrutinib and venetoclax has become an unmet urgent challenge for MCL patients. It has been shown that MCL cells developing resistance to ibrutinib through kinome-adaptive reprogramming mechanism[50]. In this study, we examined, at single-cell resolution, the cellular and transcriptomic ITH of a refractory MCL and characterized the underlying molecular cues. We demonstrated that therapeutic resistance likely arises from the high complexity of ITH and clonal evolution under therapeutic pressure; therefore, dissection of ITH and clonal evolution is critical for understanding the underlying mechanisms of therapeutic resistance and for the development of tailored treatments or precision medicine strategies to overcome these resistances.

The longitudinal sampling strategy allowed us to investigate temporal transcriptome evolution of tumor and TME cells during treatments, which represents a major advantage over single-time point studies that only capture a single "snapshot" of the evolutionary process. Indeed, although our patient population was clinically heterogeneous, greater ITH of the tumor cells and TME was observed in the nonresponders in comparison to the responders, suggesting a common mechanism underlying therapeutic resistance. For example, as an ibrutinib-venetoclax non-responder, Patient B was clearly separated from the ibrutinib-sensitive patients, particularly in the manifestation of cancer hallmarks such as OXPHOS, mTORC1, G2/M checkpoint, and MYC. However, these analyses were limited due to a small sample size. We, therefore, performed multi-platform validation of key findings at genomic and cellular levels in larger patient cohorts and also in PDX models. Recently, we functionally validated the importance of OXPHOS in ibrutinib resistance and showed that targeting OXPHOS induces pronounced anti-MCL activity in ibrutinib-resistant PDX models[15]. In contrast, these signaling pathways were gradually downregulated in response to ibrutinib treatment in the ibrutinib-responsive patient V. In addition, the single-cell analysis revealed the co-existence of molecularly distinct subpopulations in a progressive tumor (e.g., the two distinct subpopulations, B4a and B4b, identified in progressive tumor B4), which was faithfully recapitulated in the tumor-derived PDX models. These observations suggest a high degree of cellular and functional ITH in fostering the survival and clonal proliferation of resistant tumor cells, indicating that rational combinatorial therapies should be investigated to fully eradicate tumors.

Moreover, this work identified 17q gain in the nonresponders and linked 17q gain to ibrutinib resistance through integrated analysis of additional ibrutinib-resistant MCL clinical samples and cell lines. Interestingly, *BIRC5*, a gene located at 17q and encodes survivin, was remarkably upregulated in the resistant tumor cells; its association with tumor cell proliferation property; makes it a potential target for refractory MCL. YM155, the small molecule survivin inhibitor, was highly potent in targeting MCL cell lines in vitro and demonstrated significant anti-tumor activity in an aggressive dual resistant B4-derived PDX model. This demonstrates that single-cell transcriptomic analysis can identify previously undiscovered targets which has important implications for therapeutic development and precision medicine for MCL. Survivin overexpression has been identified in MCL patients in general[51], however, the survivin upregulation has never been demonstrated to contribute therapeutic resistance in MCL. Targeting survivin by the inhibitor YM155 showed potent anti-MCL activity to overcome ibrutinib-venetoclax dual resistance in vitro and in vivo. This indicates that survivin inhibition could be an alternative approach to be further investigated to overcome ibrutinib-venetoclax dual resistance in MCL and possibly other survivin-dependent malignancies.

An interesting vignette is the deep molecular profiling of the distinct cellular, transcriptomic and genomic features of ibrutinib-induced lymphocytosis and tumor clonal evolution. We observed various genomically and transcriptomically distinct cell populations released into the PB during ibrutinib treatment in the

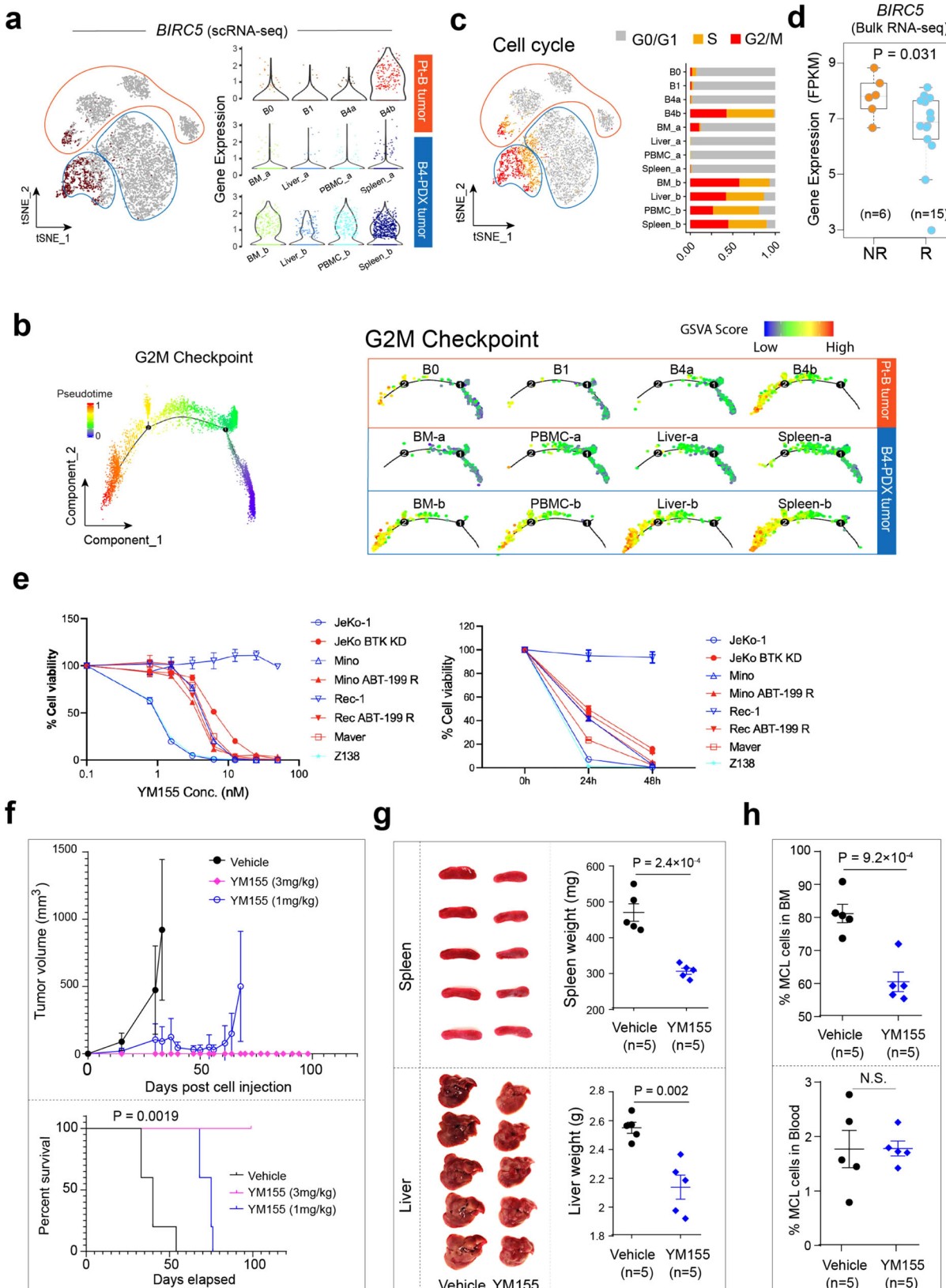

responders, leading to more effective clearance of the tumor cell populations present in blood circulation and multiple tissue compartments. Additionally, we profiled therapy-induced clonal evolution in the nonresponder patients and characterized distinct features from ibrutinib-induced lymphocytosis, e.g., the clonal expansion of tumor subpopulations that carried the driver

alterations during disease progression. Of note, the vast majority of tumor cells were cleared from PB at timepoint V6 (Fig. 1d), but with scRNA-seq, we were able to detect a single tumor cell in sample V6 and this tumor cell harbored the *CCND1* E36K mutation, a mutation that associated with ibrutinib resistance[52]. This further highlights the sensitivity of single-cell sequencing

**Fig. 5 Validation of key cancer hallmarks in the B4-derived PDX model and identification of survivin (_BIRC5_) as a target to overcome ibrutinib-venetoclax resistance. a** Feature plot of _BIRC5_ expression in cells from patient B tumor and B4-derived PDX tumors (same t-SNE plot as in Fig. 2D) (left). BIRC5 expression with violin plot (right). **b** Cells from panel A projected to a two-dimensional space by Monocle2. Each point corresponds to a single cell and cells are colored according to the inferred pseudotime (blue to red). Monocle2 was run with default parameters on the hallmark gene sets G2M Checkpoint downloaded from MSigDB. **c** Feature plot showing the cell cycle stage of each cell inferred by Seurat based on canonical cell cycle-related markers (left) and the relative proportion of cell cycle phase of cells from patient B tumor and B4-derived PDX tumors (right). **d** Differential _BIRC5_ expression via bulk RNA-seq comparing ibrutinib responders ($n = 15$) and nonresponders ($n = 6$) in a separate MCL patient cohort. The line in the box is the median value. The bottom and top of the box are the 25th and 75th percentiles of the sample. The bottom and top of the whiskers are the minimum and maximum values of the sample. _p_ value corresponds to the two-sided Wilcoxon signed-rank test. **e** The in vitro efficacy of survivin inhibitor YM155 in MCL cell lines. YM155-induced cell toxicity in MCL cell lines (red: ibrutinib-resistant; blue: ibrutinib-sensitive) in a dose (left)- and time (right)-dependent manner. The experiments were performed in triplicate ($n = 3$). Error bars represent the standard deviation (SD). **f** Mice ($n = 5$ per group) were injected subcutaneously with freshly isolated B4-PDX cells and allowed for engraftment until the tumors became palpable. The mice were then treated with continuous infusion of YM155 at 0, 1.0 or 3.0 mg/kg for 28 days. Mice were sacrificed when tumor size reached 15 mm or at day 99 post cell inoculation as end point of experiment. Plots representing tumor volume (top) and survival curves (bottom) of control and YM155-treated mice. Error bars represent the standard deviation (SD). The log-rank test was used for survival analysis. **g** Images and weights of mouse spleens and livers from B4-PDX mice model treated with vehicle or YM155. Error bars represent the standard deviation (SD). **h** The proportion of MCL cells (hCD5$^+$hCD20$^+$) in mouse BM and PM disseminations in response to YM155 ($n = 5$) compared to the control vehicle ($n = 5$). The two-sided Student _t_ test was used for statistical analysis in (**g**) and (**h**). Error bars represent the standard deviation (SD).

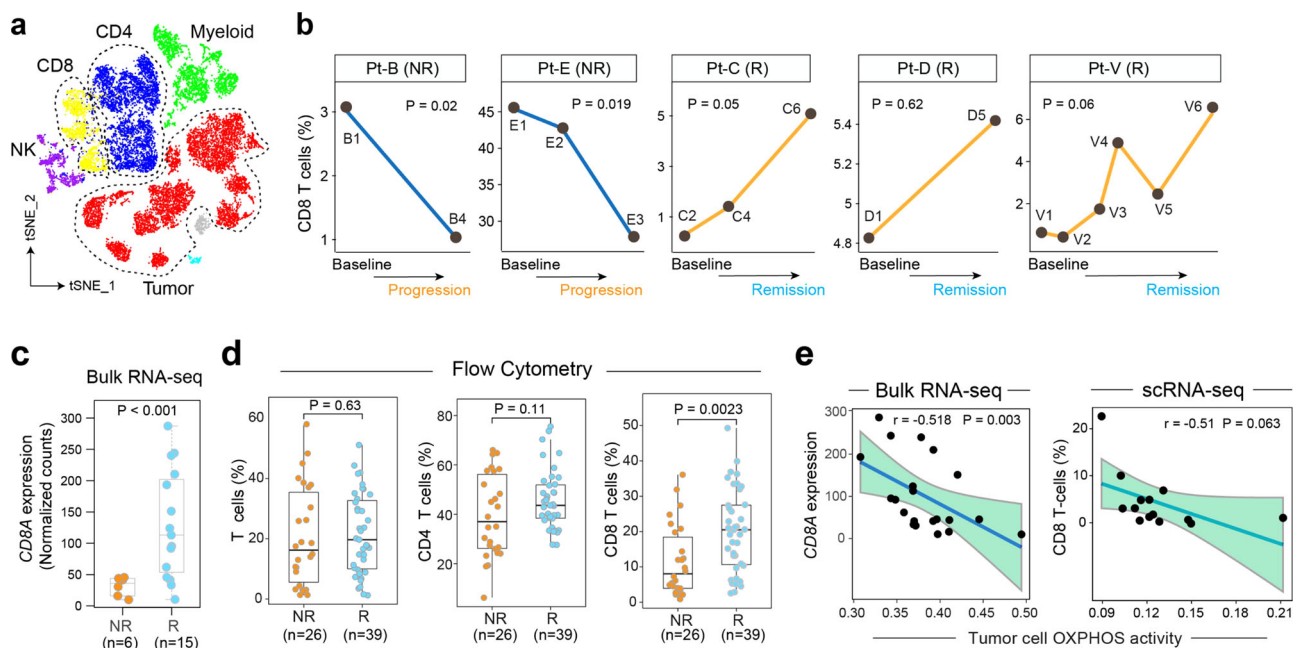

**Fig. 6 Tumor immune microenvironment diversity and evolution associated with therapeutic resistance. a** A t-SNE overview of the immune cells that passed quality control. Cells are color-coded by the defined cell types. **b** The dynamics of CD8 T cell proportion during treatment in responders (Rs) and non-responders (NRs). _p_ Values estimated by the linear regression model. **c** Differential _CD8A_ (CD8 T cell marker) expression via bulk RNA-seq comparing ibrutinib responders ($n = 15$) and non-responders ($n = 6$) in a separate MCL patient cohort. $p = 8.6 \times 10^{-4}$ from two-sided Wilcoxon signed-rank test. **d** Additional patient cohort validation using flow cytometry showing a decreased CD8+ T cell population in ibrutinib-resistant patients compared to ibrutinib-sensitive patients ($n = 65$ samples, collected from 22 patients). In **c** and **d**, the line in the box is the median value. The bottom and top of the box are the 25th and 75th percentiles of the sample. The bottom and top of the whiskers are the minimum and maximum values of the sample. _p_ Values correspond to two-side Wilcoxon Signed-rank Test. **e** Reverse correlation between CD8A expression or CD8+ T cell (%), and the tumor cell OXPHOS activity assessed by scRNA-seq. The Pearson correlation coefficient (_r_) is shown. The bounds of shape correspond to 95% confidence band for the regression line. _p_ Values in **b** and **e** correspond to F test of linear regression model.

technology, suggests the presence of potential residual disease and a necessary monitoring approach.

Lastly, we profiled the dynamic TME and tumor-immune interactions at a single-cell resolution, and our analyses demonstrated the complexity of the tumor ecosystem and a high degree of inter-tumoral cellular heterogeneity of the TME. We found that _CD8_ T-cell dysfunction likely plays a role in therapeutic resistance and potential targets are identified by analyzing aberrant cell-to-cell interaction networks. Further investigation and validation is needed to understand how OXPHOS activation in MCL tumor cells suppresses _CD8_ T-cell function and the

detailed mechanisms underlying failed immune surveillance. This work provides a framework for the discovery of new therapeutic options for therapeutically resistant MCL, potentially influencing patient care, and it provides a foundation for understanding the cellular and molecular interplay mediating therapeutic resistance.

## Methods

**Patients and patient sample collection**. Twenty-one patient samples were collected from PB, BM, or apheresis after obtaining written informed consent and approval from the Institutional Review Board at The University of Texas MD

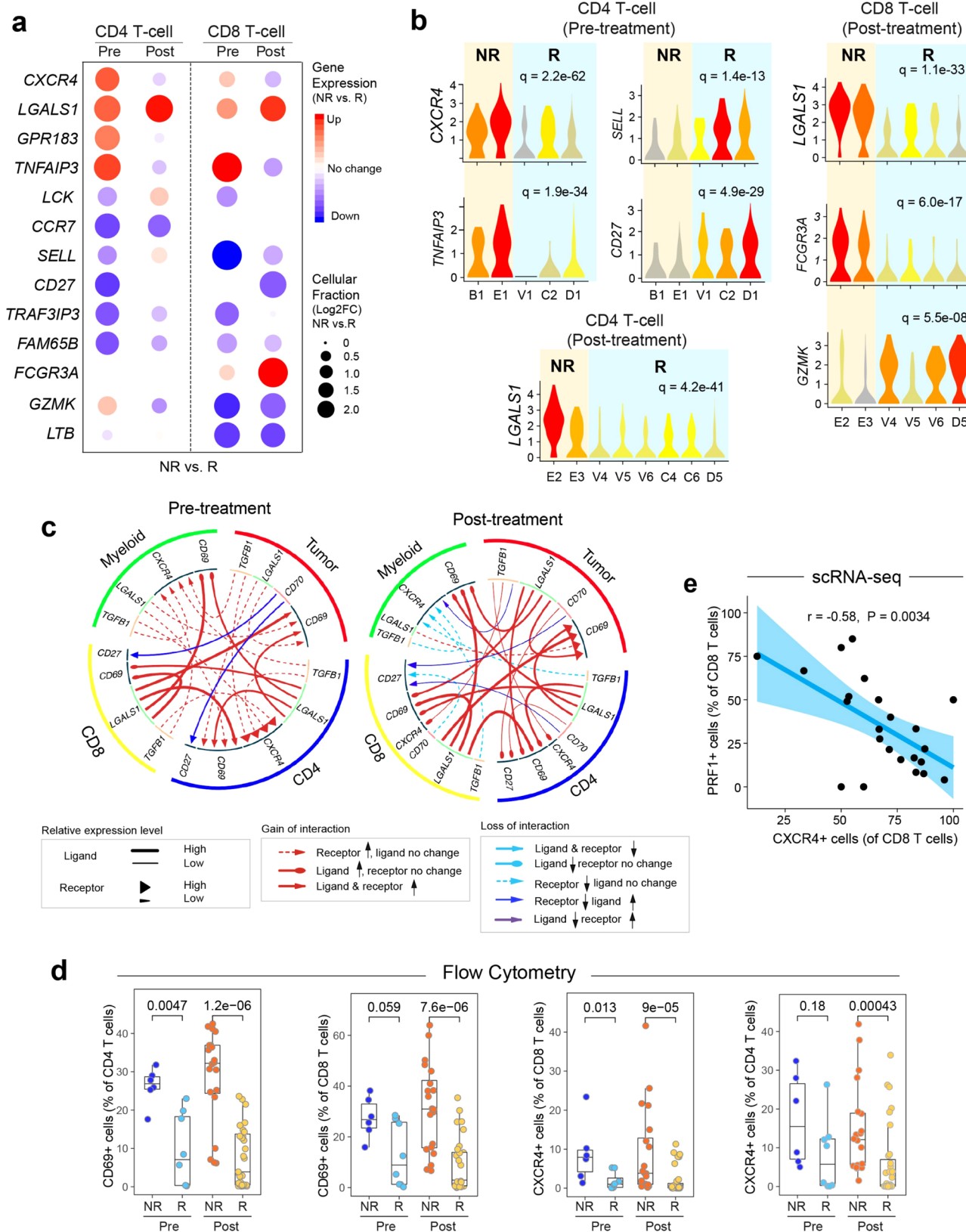

Anderson Cancer Center. The patient samples were purified by Ficoll–Hypaque density gradient centrifugation and cryopreserved before processing for scRNA-seq.

**PET/CT for MCL patients**. All MCL patient scans were acquired using the Philips C-PET System. Patients fasted for at least 6 h before the [18]F-FDG injection. The serum glucose level was determined at the time of [18]F-FDG injection using a glucometer, and all patients had glucose levels less than 130 mg/dl. Sixty-to-ninety minutes after

intravenous administration of [18]F-FDG (0.045 mCi/kg with a maximum of 6 mCi for C-PET unit and 0.14 mCi/kg with a maximum of 15 mCi for the PET/CT units), a PET/CT imaging study from the skull base to the upper thigh was acquired. Images were reconstructed by the iterative algorithm (ordered subset expectation maximization) with and without attenuation correction. A clinical report was issued at the time of performance of each scan. For the purposes of this study, all PET scan images and reports were reviewed again by a nuclear medicine specialist physician to ensure consistency of interpretation and reporting of results.

**Fig. 7 Aberrant cell-to-cell communication signaling associated with therapeutic resistance. a** Differentially expressed genes (NR vs. R) in CD4+ and CD8+ T-cells pre- and post-ibrutinib treatment, respectively. Filled circle sizes are proportional to the Log2-scaled fold changes of each gene. Upregulated genes are shown in red; downregulated genes are shown in blue. Pre: pre-treatment; Post: post-treatment. **b** Representative genes are shown in violin plots. **c** Alterations (NR vs. R) of ligand-receptor-based cell-to-cell communication networks based in pre- and post-treatment samples. **d** Flow cytometry validation of upregulated *CD69* and *CXCR4* expression in ibrutinib nonresponders in comparison to the responders in additional patient cohorts ($n =$ 65 samples collected from 22 patients). The line in the box is the median value. The bottom and top of the box are the 25th and 75th percentiles of the sample. The bottom and top of the whiskers are the minimum and maximum values of the sample. *p* Values from the two-side Wilcoxon Signed-rank Test are shown. **e** Reverse correlation between the proportion of PRF1+ CD8 T cells (cytotoxic) and the expression of CXCR4 using scRNA-seq. The bounds of shape correspond to 95% confidence band for the regression line. The Pearson correlation coefficient (*r*) is shown. *p* Value corresponds to F test of linear regression model.

**scRNA-seq library preparation and sequencing**. Single cell suspensions were isolated by Ficoll-Paque Plus (17144002, GE Healthcare Life Sciences, Pittsburgh, PA). Chromium™ Single Cell 3′ Reagent Kits v2 (PN-120237, 10× GENOMICS) were used to perform single-cell separation, cDNA amplification and library construction following the manufacturer's guidelines. A high-sensitivity dsDNA Qubit kit was used to quantify the cDNA concentration. The HS DNA Bioanalyzer for cDNA (or lower concentrated libraries) or the DNA 1000 Bioanalyzer was used to measure the concentration of the libraries. The barcoded library at the concentration of 1.6 pm was sequenced on the NextSeq500 v2.5, High Output flow cell using a 26 × 124 sequencing run format with 8 bp index (read 1).

The cellular suspensions were loaded on a 10× Chromium Single Cell Controller to generate single-cell Gel Bead-in-Emulsions (GEMs). The scRNA-Seq libraries were constructed using the Chromium Single Cell 3′ Library & Gel Bead Kit v3 (P/N 1000092, 10x Genomics). The barcoded libraries were quantified and then sequenced using the HiSeq4000 System (Illumina, San Diego, CA).

**scRNA-seq data processing and analysis**
*Raw sequencing data processing, QC, data filtering and normalization.* The raw sequencing data were preprocessed (demultiplexed cellular barcodes, read alignment, and generation of gene count matrix) using Cell Ranger Single Cell Software Suite provided by 10× Genomics. Detailed QC metrics were generated and evaluated. Genes detected in <0.1% of total sequenced cells and cells where <200 genes had nonzero counts were filtered out and not included in the analysis. Low quality cells where >10% of the counts were derived from the mitochondrial genome were also discarded. Cells with detected genes >6000 were discarded to remove likely doublet or multiplet captures. Of 20,004 sequenced cells, 18,794 (94%) passed quality filtering with an average of ~73,727 reads aligned per cell. Possible batch effects were then evaluated and corrected using FastMNN[53] with default parameters ($d = 50$, $k = 20$). Seurat was applied to the filtered matrix to obtain the normalized count as previously described[54].

*Dimensionality reduction, unsupervised cell clustering, determination of major cell types and cell states.* Seurat[55] was applied to identify highly variable genes for unsupervised clustering. The first 15 Principal Components and the top 9,305 highly variable genes were used for clustering at a resolution of 0.6. The t-distributed stochastic neighbor embedding (t-SNE) method was used for dimensionality reduction and 2-D visualization of the single-cell clusters. Feature plots were generated of suggested cell-lineage specific markers, and differentially expressed genes (DEGs) were identified using Seurat[55], followed by a manual review process to determine major cell types and cell states according to the enrichment of specific markers in each cell cluster, as previously described[56].

In addition, we applied SC3[30] a different approach for unsupervised single-cell consensus clustering analysis[57]. The SC3's parameters k, which was used in the k-means and hierarchical clustering, was chosen from 2 to 8 iteratively. For each SC3 run, the silhouette was calculated, the consensus matrix plotted, and cluster-specific genes identified. All these three aspects aid us to empirically determine the optimal k and n. Once the stable clusters were determined, the above procedure was iteratively applied to each of these clusters to reveal highly variable genes among cells in each cluster, and then use these variable genes to identify subclusters. A cutoff of adjusted *p* value of <0.01 and Auroc value of >0.6 was applied to identify most significant DEGs between SC3 clusters. inferCNV was applied to infer the CNV from scRNA-seq data (inferCNV of the Trinity CTAT Project; https://github.com/broadinstitute/infercnv). Malignant B cells were distinguished from normal B cells based on the genomic CNVs, inferred aneuploidy status, and cluster distribution of the cell.

Inferring cell cycle stage, building single-cell trajectory, pathway enrichment, and characterization of cell-to-cell communication networks. The cell cycle stage was computationally assigned for each individual cell using the R code implemented in Seurat[55] as previously described[58]. The Monocle 2 algorithm[59] was used for single-cell trajectory analysis to order tumor cells in pseudotime to infer their developmental trajectories. Monocle 2[59] was run with default parameters on highly variable genes identified by Seurat[55] and on the hallmark gene sets (MYC, OXPHOS, mTORC1, cell cycle, and PI3K/AKT/mTOR signaling) downloaded from the Molecular Signature Database (MSigDB). In order to check the robustness

of pseudo time inference, we used three algorithms TSCAN[60], Slingshot[61], and SCORPIUS[62] which are specifically designed for pseudotime inference. We, respectively, run these three algorithms with default parameters based on the same hallmark gene sets (MYC, OXPHOS, mTORC1, and cell cycle). The GVSA software package[63] was applied to identify key signaling pathways that related to ibrutinib resistance. The iTALK tool[64] was applied to characterize cell-cell communication signaling networks. The built-in database of the iTALK tool[64] was used to functionally annotate identified ligand-receptor pairs, and the visualization tool was used to generate circos plots.

**Cell culture**. The human MCL cell lines JeKo-1, JeKo BTK KD, Mino, Mino ABT-199 R, Rec-1, Rec ABT-199 R, and Maver-1 cells were maintained within RPMI 1640 medium supplemented with 1% penicillin/streptomycin, 25 mM 4-(2-hydroxyethyl)-1piperazineethanesulfonic acid (HEPES), and 10% fetal bovine serum (FBS; Sigma-Aldrich, St Louis, MO). These cells were cultured in a $CO_2$ incubator at 37 °C. The MCL cell lines Rec-1, JeKo-1, Z-138, Maver-1, JVM-2, and JVM-13 were obtained from the American Type Culture Collection (ATCC). The Mino cell line was originally established and provided by Dr. Richard Ford at MD Anderson Cancer Center. The JeKo-BTK KD cell line was generated by the MD Anderson Core Facility and previously verified and published[65]. Venetoclax (ABT-199)-resistant MCL cell lines (Mino-ABT-199 R and Rec-1 ABT-199 R) were generated from the parental cell lines (Mino and Rec-1) by multistep exposures of cells to increasing doses (up to 100 nM) of venetoclax for 8 weeks as previously described[66].

**Cell viability assay**. Cells were seeded at 10,000 cells per well in 96-well plates and treated with various doses of the indicated compounds in triplicate for 72 h and lysed with CellTiter-Glo Luminescent Cell Viability Assay Reagent (Promega, Madison, WI, USA). The luminescence was quantified using the BioTek synergy HTX Multi-mode microplate reader. The experiments were repeated at least three times. Ibrutinib (S2680) and YM155 (S1130) were purchased from Selleck Chemicals (Houston, TX, USA).

**Apoptosis assay**. Annexin V-binding assay was used to detect apoptosis. MCL cells were seeded in 48-well plates, treated with vehicle or YM155 (50 nM) for 24 and 48 h, and were stained with Annexin-V and propidium iodide (Abcam, Cambridge, UK). Flow cytometric analysis was performed immediately with a Novocyte Flow Cytometer (ACEA Biosciences, San Diego, CA, USA) to determine the percentages of Annexin-V positive cells. Data were analyzed with NovoExpress (ACEA Biosciences, San Diego, CA, USA) or FlowJo10 (Tree Star, Ashland, OR, USA). The experiments were repeated at least three times.

*Cell cycle arrest assay.* MCL cells in triplicate treated with vehicle or YM155 (50 nM) for 24 h were fixed in 50% pre-cold ethanol and stained with propidium iodide followed by flow cytometric analysis with a NovoCyte Flow Cytometer (ACEA Biosciences) to quantify the cell cycle stages. The experiments were repeated at least three times.

**Establishment of B4-PDX model in NSG mice**. The Institutional Animal Care and Use Committee of The University of Texas MD Anderson Cancer Center approved the experimental protocols. One vial of frozen B4-derived MCL cells was thawed, and $20 × 10^6$ cells were injected into NSG mice intravenously via the tail vein. Blood collection and flow cytometry to detect MCL cells in mouse blood (>1%). The B4-derived MCL cells disseminated to the spleen, liver, BM, blood, and others, and a PDX model named B4-PDX was established. Freshly isolated PDX cells from the mouse spleen were used to pass onto the next generation through intravenous inoculation. Generations G3–G6 were used in this study to test in vivo drug efficacy in this model.

**Collection and scRNA seq analysis of B4-PDX samples**. Primary PDX cells were freshly isolated from the spleen, liver, BM and PB of one B4-PDX mouse with splenomegaly, hepatomegaly, and high involvement in the BM and PB. The freshly

isolated primary PDX cells were subject to scRNA seq and analyzed as primary patient samples.

**In vivo efficacy of YM155 in the B4-PDX model**. In cohort I, freshly isolated primary PDX cells from the mouse spleen of the B4-PDX model ($2 \times 10^6$) were injected into 6–8-week-old female NSG mice intravenously via the tail vein. The mice started the treatments at 4 weeks post cell inoculation and were treated with 2 cycles of 7-day continuous infusion with vehicle or YM155 (1.0 mg/kg) plus 2 weeks off. The mice were monitored weekly, and the PB was collected every other week before, during, and at the end of treatment. B2M production in the mouse serum was measured via ELISA. At the end of the experiment, the mice were euthanized and dissected for blood, spleen, liver, and BM. The weight of the spleen and liver was measured, and the cells from the blood, spleen, liver, and BM were isolated and stained for fluorescence-conjugated human anti-CD5 and human anti-CD20 antibodies. CD5- and CD20-double-positive cells representing the MCL cell population present in each organ or tissue were detected by flow cytometry. In cohort II, freshly isolated primary PDX cells from the mouse spleen of the B4-PDX model ($10 \times 10^6$) were injected into NSG mice subcutaneously. When the subcutaneous xenografts were palpable, the mice were continuously infused with vehicle or YM155 (1 or 3 mg/kg) for 4 weeks and off for the rest of the experimental period. The mice survival and PDX size were monitored twice a week. The tumor volume was calculated by $V = (W^2 \times L)/2$, where $V$ is tumor volume, $W$ is tumor width, and $L$ is tumor length.

**Flow cytometry analysis**. Single cell suspensions of the primary patient cells collected from patient blood or BM were used to perform 17-color flow cytometry analysis (Supplementary Fig. 11). The antibodies used in this analysis were (Supplementary Data 8): CD28-BUV395 (740308, BD Biosciences, San Jose, CA), CD127-BUV737 (612794, BD Biosciences, San Jose, CA), CD69-eFluor 450 (48-0699-42, Life technologies, Calsbad, CA), CD279-SB600 (63-2799-42, Life technologies, Carlsbad, CA), CD223-SB645 (64-2239-42, Life technologies, Callsbad, CA), CD3-BV711 (563725, BD Biosciences, San Jose, CA), CD8-SB780 (78-0086-42, Life technologies, Carlsbad, CA), LAP-Alexa Fluor 488 (FAB2463G, R&D systems Inc., Minneapolis, MN), CD4-PerCP-Cy5.5 (560650, BD Biosciences, San Jose, CA), CD272-PE (344506, Biolegend, San Diego, CA), CD197-PE-eFluor 610 (61-1979-42, Life technologies, Carlsbad, CA), CD19-PE-cy5 (555414, BD Biosciences, San Jose, CA), CD184-PE-Cyanine7 (25-9999-42, Life technologies, Carlsbad, CA), LGALS1-AF647 (sc-166618AF647, Santa Cruz Biotechnology, Dallas, TX), CD45RO-Alexa Fluor 700 (561136, BD Biosciences, San Jose, CA), CD152 (47-1529-42, Life technologies, Carlsbad, CA), and LIVE/DEAD Fixable Aqua Dead Cell Stain (L34957, Life technologies, Carlsbad, CA). The cells were washed in phosphate-buffered saline (PBS) and stained with a cocktail of cell surface antibodies and LIVE/DEAD Fixable Aqua Dead Cell Stain except for LGALS1-AF647 on ice for 30 min. The cells were then washed in PBS, fixed, permeabilized and stained with intracellular antibody LGALS1-AF647 for 20 min. The cells were washed again and resuspended in 1% paraformaldehyde fixative solution and analyzed by flow cytometry (LSRFortessa X-20 analyzer) (BD Biosciences, San Jose, CA within 24 h).

**Deep whole-exome sequencing**. Briefly, indexed libraries were prepared from 500 ng of Biorupter Ultrasonicator (Diagenode, Denville, NJ, USA)-sheared, genomic DNA using the KAPA Hyper Library Preparation Kit (KAPABiosystems, Wilmington, MA, USA). The indexed libraries were prepared for capture with six cycles of pre-ligation-mediated PCR amplification. Following amplification and reaction cleanup, the libraries were quantified using the Qubit™ dsDNA HS Assay (ThermoFisher, Waltham, MA, USA) and assessed for size distribution using the Fragment Analyzer (Advanced Analytical, Ames, IA, USA). Library concentrations were normalized, and the libraries were multiplexed 8 libraries/pool. Each multiplexed library pool was hybridized to a probe pool from the SeqCap EZ Human Exome Enrichment Kit v3.0 (Roche-NimbleGen, Madison, WI, USA). The enriched libraries were amplified with eight cycles of post-capture PCR, then assessed for exon target enrichment by qPCR. The exon-enriched libraries were then assessed for size distribution using the Fragment Analyzer (Advanced Analytical) and quantified by qPCR using the KAPA Library Quantification Kit (KAPA Biosystems). Sequencing was performed on the HiSeq4000 Sequencer (Illumina, San Diego, CA, USA), one capture (eight samples) per lane using the 150 bp paired-end configuration.

**WES data processing and genotyping quality check**. WES data was processed in a similar way as described in our recent study[15]. Raw output of the Illumina exome sequencing data was processed using Illumina's Consensus Assessment of Sequence and Variation (CASAVA) tool (v1.8.2) (http://support.illumina.com/sequencing/sequencing_software/casava.html) for demultiplexing and conversion to FASTQ format. The FASTQ files were aligned to the human reference genome (hg19) using BWA (v0.7.5)[67] allowing up to 3 mismatches (2 mismatches must be in the first 40 seed regions). The aligned BAM files were then subjected to mark duplication, realignment and base recalibration using Picard (v1.112) and GATK (v3.1-1) software tools[68]. The generated BAM files were then used for downstream analysis. Genotyping quality check was performed to rule out any possible sample swapping or contamination. Briefly, germline SNPs were called using Platypus (v0.8.1). Samples from the same patient were confirmed/identified by the percentage of genotyping-

identity between them, which was defined by the fraction of identical germline alleles among the overlapping SNPs between the two samples. All samples in this study passed quality check, and no sample swapping or contamination was detected.

**Somatic mutation calling, filtering, functional annotation, and analysis of clonal architecture**. MuTect (v1.1.4)[69] was applied to identify somatic point mutations, and Pindel (v0.2.4)[70] was applied to identify small insertion and deletions (Indels). The MuTect and Pindel outputs were then run through our pipeline for filtering and annotation. Briefly, only MuTect calls marked as "KEEP" were selected and taken into the next step. For both substitutions and Indels, mutations with a low variant allelic fraction (VAF < 0.05) or had a low total read coverage (<20 reads for tumor samples; <10 reads for germline sample), were removed. In addition, Indels that had an immediate repeat region within 25 base pairs downstream towards its 3′ region were also removed. After that, common variants reported by the ExAc (the Exome Aggregation Consortium, http://exac.broadinstitute.org), Phase-3 1000 Genome Project (http://phase3browser.1000genomes.org/Homo_sapiens/Info/Index), or the NHLBI GO Exome Sequencing Project (ESP6500, http://evs.gs.washington.edu/EVS/) with a population minor allele frequency greater than 0.5% were removed. The intronic mutations, mutations at 3′ or 5′ UTR or UTR flanking regions, silent mutations, small in-frame insertions and deletions were also removed. SciClone[71] was applied to somatic mutation data to infer clonal architectures and tumor evolution patterns, and R-fishplot[72] was used for visualization.

**Estimation of total copy numbers and allele-specific copy numbers**. Total DNA copy number analysis was conducted using an in-house application ExomeLyzer[73] followed by CBS segmentation[74]. The copy number segmentation files were loaded to R for visualization (e.g., Figure 3d). R package "CNTools" (v1.24.0) was used to identify copy number gains (log2 copy ratios > 0.3) or losses (log2 copy ratios < −0.3) at the gene level. We used Sequenza[75] to infer the allele-specific DNA copy number alterations. Sequenza was applied to the WES bam files of matched tumor-normal pairs from patients B and V, respectively. Sequenza outputs (e.g., B allele frequency and absolute allele-specific copy numbers) were merged and compared within and across patients and also compared with the inferred copy number profiles from the scRNA-seq platform.

*Statistics and reproducibility*. In addition to the bioinformatics approaches described above for scRNA-seq data analysis, all other statistical analysis were performed using statistical software R v3.4.3. Analysis of differences in cancer hallmarks and immunological features (continuous variables) between response (R vs. NR) groups was determined by the nonparametric Mann–Whitney $U$ test. To control for multiple hypothesis testing, we applied the Benjamini–Hochberg method to correct $p$ values and calculated the false discovery rates ($q$-values). For functional experiments, all assays were performed in triplicate and expressed as mean + SD. Statistical analysis was performed using GraphPad Prism v8.00 software. Statistical significance was determined by the two-sided two-sample $t$ test analysis. Violin plots were generated using the geom-violin from the ggplot2 R package.

**Reporting summary**. Further information on research design is available in the Nature Research Reporting Summary linked to this article.

## Data availability
WES and All single-cell expression data generated in this study have been deposited in the European Genome-Phenome Archive (EGA) database under the accession code EGAS00001005019. Request for the relevant data can be made to the corresponding author. The genotyping data is available within this paper and the supporting information files. In addition to the datasets generated internally for this study, we downloaded the normalized expression data of two MCL patient cohorts under the access code GSE10793[43] and the website: https://llmpp.nih.gov/MCL/ [https://llmpp.nih.gov/MCL/proliferation_signature_expression.txt][44], in order to assess the prognostic significance of survivin upregulation in MCL. 50 hallmark cancer gene sets were downloaded from MSigDB (https://www.gsea-msigdb.org/gsea/msigdb/index.jsp). All other remaining data are available from the corresponding authors upon request.

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

## Acknowledgements

We thank the patients and their families who contributed to this research study. This study was supported by the generous philanthropic funds from The Gary Rogers Foundation, Kinder Foundation, and the Cullen Foundation, and the start-up research fund kindly provided to L. Wang by MD Anderson Cancer Center. This study was also supported by the NIH-funded Cancer Center Support Grant (CCSG) P30 CA016672 (Peter Pisters, Principal Investigator) and the NIH Core Grant for the Sequencing and Microarray Facility (CA016672). We thank Dr. Adam for her critical editing of the paper. The Advanced Cytometry & Sorting Core Facility is supported by NCI P30CA016672.

## Author contributions

M.W. and L.W. jointly supervised the study. V.J., L.W., and M.W., conceived the experiments. V.J. contributed to the patient sample preparation and conducted functional experiments. S.Z. contributed to the bioinformatics data analysis and integration, and generation of figures and tables for the manuscript. L.W. supervised the bioinformatics data analysis, data integration, and interpretation. V.J., J.L., G.H., D.H., M.H., R.W., M.D., E.D., Y.L., R.Z., Y.W., R.N., J.M., J.Z., S.A., C.F., X.S., and K.N. assisted with experiments and data analysis. V.J., M.W., D.S., M.B., K.H., J.B., A.L., A.R., S.T., H.H., P.J., C.Y.O., and Z.C. contributed to the sample collection, processing, and collection of clinical data. N.W.B. provided expertise in reviewing the PET/CT images. L.W., S.Z., V.J., K.N., H.L., R.S., P.J., J.R., A.F., and M.W. wrote and revised the paper. Q.C. contributed to bioinformatics data analysis of cell lines.

## Competing interests

The authors declare no competing interests.
