## [Peer Review File · Nature Communications]

Reviewers' Comments:

Reviewer #1:

Remarks to the Author:

Zhang and coworkers present a single cell RNA-seq study in longitudinal samples of MCL patients. These data are used to shed light into the molecular heterogeneity of this disease and the mechanisms of evolution and therapy resistance. The topic is timely important and the manuscript uses state-of-the-art methods. However, after reading the manuscript I have several important concerns that make me doubt about the relevance of the results to better understand MCL pathobiology and clinical behavior in the context of drug resistance. I hope these comments may help the authors to improve your manuscript.

Major comments:

1. Frankly speaking, my main concern is the study design itself. The authors recruited for this study samples from 5 patients, from which they have 3-8 time points, related to different steps of disease progression, i.e. diagnosis, treatment and possible relapse. Three of these patients were classified as responsive and two as non-responsive. However, the majority of conclusions presented in this manuscript are drawn from a deep analysis of only one patient, the non-responsive case B, and as such, it looks like a "case report" with a limited scope. I am not questioning that this case is indeed very interesting, but the authors should be cautious in drawing general conclusions about the molecular mechanisms of drug resistance in MCL, if in fact they present just a case study.
2. Continuing with the study design, the authors introduce the case of patient E as non-responsive MCL. However, as shown in Fig1d, this patient hardly has any B cells! It is hard to understand how a B cell tumor that is not responding to therapy does not show any increased number of B cells in any sample. This is inconsistent with the flow cytometry data presented in Supplementary Table 1, where the authors claim to detect a CD19+ and CD20+ fraction in this case. It is also puzzling that no Cyclin D1 translocation can be detected. This may happen in very rare MCL cases, as there might be some cryptic rearrangements leading to CCND1 overexpression. Overall, the evidence shown in the manuscript makes me doubt about the MCL diagnosis in this case.
3. As described in Methods section, the patient samples were collected from different sources, i.e. peripheral blood, bone marrow or apheresis. The transcriptome from different microenvironments, especially at the single-cell level, may represent some important differences with some specific cell subpopulation that may be specific for one source or for the other. Did the authors take this issue into consideration? How can the reader be sure that the differences among cell subpopulations in the different cases (i.e. Fig1 or Supplementary Fig1 and also TME studies in Figure 6) or even differences in the selected genes expression are not biased by different sources?
4. I am definitely missing an effort to perform some more overall analysis in order to explain transcriptome dynamics common to all MCLs. That could be made at least for non-responsive patients, i.e. C, D and V. I believe this may give some interesting message about the changes in gene expression and clones composition upon ibrutinib treatment. Instead, the authors are grounding some of their analysis (i.e. Fig5, analysis of pathways) on the comparison between non-responsive patient E and only one responsive patient V, that they define as "representative". What is the rationale behind selecting this particular case as an example of responsive MCL? Again, this study seems to draw general conclusions from a kind of case report study design.
5. The role of 17q gain in the refractory MCL sample, acquired in B4 clones, might be indeed interesting. The authors identified a list of 55 genes that were significantly overexpressed in B4 versus all other samples (both from B patient and other analyzed patients). However, analyzing scRNA-Seq heatmap (Fig.3e) as well as Supplementary Table 2, it seems that there might be some differences within subpopulations of B4 samples (for example between B4a and B4b). Therefore, there might be some other differences between these two subpopulations that are not necessarily related to 17q gain. Furthermore, if I understood properly the data shown in Suppl.Table 2, only 22% of all cells from B4 samples showed overexpression of BIRC5, thus the link of this gene and 17q gain in non-responsive fraction does not seem that strong. In the line of this observation, the validation of BIRC5 expression levels in an additional series of non-responsive and responsive MCL

samples are borderline and not very robust. It looks to me that the reported difference is due to the low expression (approx. 3 FPKM) of one outlier from resistant MCLs. Taken together, I believe that the authors should be more careful in drawing a general conclusion about the role of 17q gain and in particular of BIRC5 in the responsiveness to treatment.

6. In relation to the above-mentioned studies of BIRC5, I found it difficult to understand why the Z138 cell line was not included in the cell line panel to test survivin inhibitor YM155. As authors mentioned previously, they found 17q gain only in this cell line, thus this should be the best model to check their hypothesis.

7. The single-cell techniques give a great opportunity to study clonal dynamics in the course of disease progression and theoretically, to detect even minor subpopulation that may arise at different time points. Unfortunately, I feel that the authors did not exploit sufficiently the potential of the data that they've generated. For instance, sometimes different subpopulation don't have to form a completely separated "cloud" in t-SNE plot, but may be distinguished by different level of expression of some genes. This kind of analysis would give a lot of potentially interesting information about heterogeneous landscape of MCL patients at different time point. Although I recognize that the authors have made some effort to track the clonal evolution, it was done in a superficial way and only in two patients. A more profound analysis of clonal evolution could be performed.

8. The discussion contains passages of overinterpretation, and the results should be interpreted and discussed within the limitations of the study design, that should be recognized and openly discussed.

Minor issues:

-MCLs are classified into conventional and leukemic non-nodal according to the WHO classification. Can the authors provide the SOX11 status and IGHV somatic hypermutation of the patients. A better description of clinico-biological features of the patients would be important.

- Why heatmap 3e shows only around 18 genes? The authors mention that they determined 55 differential genes.

- The color code is not helping in understanding several figures, e.g. Suppl. Fig1a, headings in the heatmaps in Suppl. Fig2. Please make sure the color codes clarify rather than confuse the reader.

Reviewer #2:

Remarks to the Author:

MCL is an aggressive B-cell lymphoma with poor prognosis due to emergence of drug resistant populations and lymphoma progression. Ibrutinib, a new FDA-approved Bruton's tyrosine kinase (BTK) inhibitor, was shown to have high response rates in MCL patients. However, as the use of this drug continues to grow in MCL and other B-cell lymphomas, emergence of drug resistance and fatal progression are of increasing clinical concern. Remarkably, once MCL patients relapse after or on ibrutinib treatment there is rapid disease progression and patients die within 12 months. Thus, there is an urgent need to define mechanisms of ibrutinib resistance (IR) and to identify novel targets to bring forward novel treatment options with real curative potential for this fatal complication. Given that the mechanisms driving IR are poorly understood and no recurrent driver mutations have been identified in MCL, Zhang et al modeled IR evolution mechanism by implementing single-cell RNA sequencing in IR MCL lines and primary samples. They characterized MCL molecular heterogeneity and immune cellular diversity that drive drug resistance evolution and addressed an important clinical topic.

First of all, by using ibrutinib responders (n=3), non-responders (2) and control normal B-lymphocytes (2), scRNA-seq and WES, Zhang et al characterized molecular and cellular heterogeneity by patient, by response and cell type, and revealed that multiple cancer hallmark pathways (Myc, oxphos and mTOR) and acquisition of 17q were associated with IR evolution. These findings were further validated at genomic and cellular levels in extended primary patient

samples and PDX models. Functionally, they showed that BIRC5/survivin, amplicon at 17q, upregulated and regulated cell survival and growth ex vivo and in vivo in resistant MCL tumor cells. Second, they explored the TME immunity in IR MCL samples and discovered significant different dynamics of CD8 T lymphocytes during ibrutinib treatment in ibrutinib responsive and IR patients and these changes were contributed to alteration of CD69 and CXCR4 expression. Overall the data show an interesting mechanism by which intratumor heterogeneity and TME immune cell dysregulation contribute to drug resistance evolution. Finally, the functional significance ex vivo experiments was validated by in vivo PDX MCL xenografts and primary samples. The paper is generally well presented with strong clinical correlation and bioinformatics analysis. Overall, the data shown are robust, convincing and the experiments well carried out.

However, to address some minor points will strength the manuscript.

In discussion, a short explanation should be added to emphasize that, in addition to intratumor heterogeneity, tumor cell plasticity (transcriptome and kinome reprogramming) also play a critical role in IR evolution. Also, discussion on potential treatment options such as epigenetic modulation to prevent the onset of drug resistance.

Reviewer #3:

Remarks to the Author:

Summary: In this manuscript, the authors performed longitudinal scRNA-seq analysis on PBMCs of mantle cell lymphoma (MCL) patients. They identified that tumor B cells in an ibrutinib non-responsive (NR) patient have unique cancer hallmarks distinct from that in responsive (R) patients, which was validated using a PDX mouse model. Furthermore, NR tumor B cells have 17q gain which induces the upregulation of survivin in NR patients compared to responsive patients. The authors also demonstrated that targeting survivin with YM155 is an effective therapeutic approach for MCL. Finally, they characterized differences in the tumor microenvironment between R and NR MCL patients.

The present manuscript is potentially important because it provides a potential mechanism of ibrutinib resistance in MCL. However, their claims are not convincingly supported by data and cannot be generalized since only one NR patient was considered.

Major points:

1. Figure 1c and 6a: The normal cell clusters of MCL patients are separated from that of healthy controls. The authors argued this suggests TME reprogramming, but the possibility of batch effects between MCL patients and healthy controls cannot be excluded. The authors used unusual high number of highly variable genes (9305) for clustering and dimensionality reduction, which might be a cause of the strong batch effects between MCL patients and healthy controls. I'm wondering whether reducing the number of highly variable genes or using batch correction methods resolves this issue.

2. Figure 1d: In Patient E (NR), the fraction of tumor B cells is negligible and not detected in E2, which is not consistent with the clinical data presented in Figure 1b and the argument stating that the fraction of tumor B cells in NR patients increases during treatment. Since the authors used the cryopreserved samples, it might introduce selective cell losses. Most findings were derived from one NR patient (Patient B), which is difficult to generalize and the main weak point in this manuscript.

3. Figure 2d-f: The two cell subclusters of B4-PDX tumor cells should be indicated in Figure 2d,f and overlaid in Figure 2e. The robustness of trajectory analysis should be validated by using another pseudotime inference method.

4. Figure 3a,b: The normal cells should be included as a control. It seems that a subset of tumor B

cells have normal CNV profiles, indicating that they might be normal B cells in MCL patients. This issue should be carefully examined. In Figure 3a, the CNV profiles of B4a and B4b look similar. However, they are segregated by the CNV profiles in Figure 3b. Which features in the CNV profiles determine the separation?

5. Figure 4b: Do the six NR patients have 17q gain compared to 15 R patients?

6. Figure 4e: It seems that there is no difference of YM155 in vitro efficacy between ibrutinib-resistant and sensitive MCL cell lines, suggesting that YM155 is not specific to ibrutinib-resistant tumor B cells. Why is Rec-1 not responsive to YM155? Is survivin lowly expressed in Rec-1? How about the ibrutinib resistant Z-138 in Figure 2d? Do the ibrutinib-resistant cell lines have 17q gain and survivin overexpression compared to the sensitive cell lines? Does the combinatorial treatment of ibrutinib and YM155 increase the survival rate of MCL patients by targeting both sensitive and resistant clones?

7. Figure 5e: The authors argued that V3/V4 tumor cells are new subpopulations redistributed from spleen, which are distinct from V0/1/2. However, V5 tumor cells, which should be more similar to V3/4, are clustered together with V0/1/2, which is contradictory to clonal evolution.

8. Figure 6e: Are there any other gene set signature correlated with the proportion of CD8+ T cells?

Minor points:

1. What is the cell type of the skyblue colored cluster in the bottom of Figure 1c?

2. Line 150 "Overall, 13 cancer hallmarks were significantly upregulated in the ibrutinib-resistant tumors": The authors should report the statistical significance of this sentence.

3. Line 157-158 "significantly upregulated BIRC3": Report the statistical significance.

4. Figure 2e, 4c, S2f: The corresponding gene set signature score should be also overlaid on the Monocle2 plot for better visualization.

5. Line 243: "high surviving expression highly" \diamond "high surviving expression IS highly"

6. Line 254: "B2M checkpoint" \diamond "G2M checkpoint"

7. Figure 4d: The cell-cycle stage fraction of B4-PDX cells should be also shown for minor and major subclusters defined in Figure 2d.

8. Line 277: Define "B2M".

9. Line 345 "we observed a significant decrease": Report the statistical significance.

Dear Reviewers,

We would like to express our sincerest gratitude to all of you for your insightful reviews and constructive comments on our manuscript (NCOMMS-20-06359-T). Your insights and queries have helped us to significantly strengthen our manuscript. The manuscript has been substantially revised based on your reviews. Point-by-point responses to your comments are listed below:

Response to Reviewer #1's comments: Page 2

Response to Reviewer #2's comments: Page 11

Response to Reviewer #3's comments: Page 12

We inserted the newly added figures and contents into the letter for easy accessibility. We apologize for the long letter.

Reviewers' comments:

Reviewer #1 (Remarks to the Author): Expert in lymphoma

Zhang and coworkers present a single cell RNA-seq study in longitudinal samples of MCL patients. These data are used to shed light into the molecular heterogeneity of this disease and the mechanisms of evolution and therapy resistance. The topic is timely important and the manuscript uses state-of-the-art methods. However, after reading the manuscript I have several important concerns that make me doubt about the relevance of the results to better understand MCL pathobiology and clinical behavior in the context of drug resistance. I hope these comments may help the authors to improve your manuscript.

Major comments:

1. Frankly speaking, my main concern is the study design itself. The authors recruited for this study samples from 5 patients, from which they have 3-8 time points, related to different steps of disease progression, i.e. diagnosis, treatment and possible relapse. Three of these patients were classified as responsive and two as non-responsive. However, the majority of conclusions presented in this manuscript are drawn from a deep analysis of only one patient, the non-responsive case B, and as such, it looks like a “case report” with a limited scope. I am not questioning that this case is indeed very interesting, but the authors should be cautious in drawing general conclusions about the molecular mechanisms of drug resistance in MCL, if in fact they present just a case study.

We thank the reviewer and certainly agree with this important comment. We have reframed our manuscript along the lines of a case report for patient B, still keeping other patients though at this time, as suggested by the editor. In addition, we have included language in the Discussion section on the limitation of this study.

Discussion, Page 21, Line 480-483:

“...However, these analyses were limited due to a small sample size. We therefore performed multi-platform validation of key findings at genomic and cellular levels in larger patient cohorts and also in PDX models...”

2. Continuing with the study design, the authors introduce the case of patient E as non-responsive MCL. However, as shown in Fig1d, this patient hardly has any B cells! It is hard to understand how a B cell tumor that is not responding to therapy does not show any increased number of B cells in any sample. This is inconsistent with the flow cytometry data presented in Supplementary Table 1, where the authors claim to detect a CD19+ and CD20+ fraction in this case. It is also puzzling that no Cyclin D1 translocation can be detected. This may happen in very rare MCL cases, as there might be some cryptic rearrangements leading to CCND1 overexpression. Overall, the evidence shown in the manuscript makes me doubt about the MCL diagnosis in this case.

We thank the reviewer for this thoughtful comment and apologize for not having detailed description of the patient E in our original manuscript. For Single-cell RNA-seq, we

included the pre-ibrutinib PBMC sample collected from Patient E (because of no residual BM specimens available at that time point). From the baseline PBMC sample, we only captured 9 MCL cells, which were too few for subsequent analysis of tumor-intrinsic resistance factors. We therefore only used the non-malignant cells from patient E to help understand the TME related factors.

The flow cytometry data on Patient E showing 40-50% MCL cells (CD19+ and CD20+, Supplementary Table 1 in revised manuscript) was generated from a bone marrow sample. Although IGH CCND1 FISH analysis showed negative results for CCND1 translocation, IHC staining of informative markers showed positive expression of cyclin D1 in the bone marrow biopsies from this patient (Fig. R1). Flow cytometry immunophenotyping detected a monotypic B-cell population co-expressing CD5, CD19, CD20, CD22, CD38, CD44, CD79b and surface kappa light chain. The neoplastic cells are negative for CD3, CD4, CD8, CD10, CD11C, CD23, CD30, CD43, CD200 and lambda light chain. The primary diagnosis of patient E was confirmed through pathology re-reviewed and the results were verified independently by two experienced lymphoma pathologists. We included this in our revised manuscript (Page 5, Line 105-107).

Patient E

Fig. R1 CCND1 IHC Image for patient E.

3. As described in Methods section, the patient samples were collected from different sources, i.e. peripheral blood, bone marrow or apheresis. The transcriptome from different microenvironments, especially at the single-cell level, may represent some important differences with some specific cell subpopulation that may be specific for one source or for the other. Did the authors take this issue into consideration? How can the reader be sure that the differences among cell subpopulations in the different cases (i.e. Fig1 or Supplementary Fig1 and also TME studies in Figure 6) or even differences in the selected genes expression are not biased by different sources?

We thank the reviewer for bringing up an excellent point and appreciate the reviewer's concern regarding the heterogeneous sample sources. There are 15 PBMC samples, 5 BM samples and 3 apheresis samples in our analysis (Fig. R2A). We first examined whether different tissue source would be a bias for cell composition analysis. We analyzed the cell composition for each major cell types including CD4, CD8, NK, and Myeloid cells across different tissue sources, but observed no statistical difference

($P > 0.05$, Fig. R2B). As 15 out of 23 samples were PBMCs, and the cell composition of PBMC and apheresis appears similar, our original analysis included only PBMC and apheresis samples and excluded BM samples for TME (tumor microenvironment) profiling. To address the reviewer's question, we redid the analysis using only the PBMC samples (excluding apheresis and BM samples), but observed no significant impact on the cell composition dynamics (Fig. R2C). We included this analysis in revised manuscript (Page 6 Line 133-135).

Fig. R2 Evaluation of batch effect. (A) Cell composition dynamics at different time points during sample collection. (B) Tissue source composition of cell type before batch effect correction. (C) The relationship of time points with cell type proportion identified based on raw and batch effect corrected data.

4. I am definitely missing an effort to perform some more overall analysis in order to explain transcriptome dynamics common to all MCLs. That could be made at least for non-responsive patients, i.e. C, D and V. I believe this may give some interesting message about the changes in gene expression and clones composition upon ibrutinib treatment. Instead, the authors are grounding some of their analysis (i.e. Fig5, analysis of pathways) on the comparison between non-responsive patient E and only one responsive patient V, that they define as "representative". What is the rationale behind selecting this particular case as an example of responsive MCL? Again, this study seems to draw general conclusions from a kind of case report study design.

We are very thankful for this suggestion. To address this critique, we performed additional analyses.

To identify the transcriptomic features that are common to all MCL cells, we performed differential gene expression analysis between the MCL cells from samples at baseline and the normal B cells from healthy donors, and compared the overlap between sets of DEGs across the baseline samples from 5 patients. We identified 6 downregulated genes and 20 upregulated genes that were ubiquitous to all baseline samples from 5 patients (Fig. R3A and B, Supplementary Table S2 in revised manuscript). The downregulated genes included PIK3IP1 (a negative regulator of PI3K) (PMID: 18632611), and DDIT4 (an inhibitor of mTORC1 signaling) (PMID: 30745581). The upregulated genes included CCND1, STMN1 (also named oncoprotein 18, frequently expressed in high-grade lymphoma) (PMID: 8412315), MARCKS (the major protein kinase C substrate that regulates PI3K/AKT signaling) (PMID: 28166200, 27119641), FCRLA (a tumor-associated antigen of BCL) (PMID: 17625599), FCRL2 (a prognostic marker of CLL with strong correlation with mutated IGHV status) (PMID: 18314442), and VPREB3 (a pre-B-cell receptor associated protein and a diagnostic marker for identifying c-MYC translocated lymphomas) (PMID: 20823132). We note that the expression levels of STMN1 and MARCKS were significantly elevated in MCL cells from B4 at disease progression (Fig. R3C and D), suggesting a potential role of STMN1 and MARCKS in promoting MCL progression.

We include this analysis in revised manuscript (Page 6 Line 142-154).

Fig. R3 Significantly differential expressed genes in MCLs. (A) Venn plot for down-regulated and (B) up-regulated genes in MCLs. Examples for down- (C) and up-regulated (D) genes.

We agree with the reviewer that it would be interesting to investigate transcriptome dynamics common to MCLs upon ibrutinib treatment, which was done in longitudinal samples collected from patients B and V. For responders including patients C and D, tumor cells in PBMC were cleared quickly upon ibrutinib treatment and therefore, very few tumor cells were detected in PBMC from these two patients after treatment. Patient V was a slower ibrutinib responder and we were able to detect a good amount of tumor cells in samples collected during the first weeks post ibrutinib treatment (V2, V3, V4 and V5, but not V6). Following your suggestion, we have reframed our manuscript along the lines of a case report.

Reference

1. He, X. et al. PIK3IP1, a negative regulator of PI3K, suppresses the development of hepatocellular carcinoma. *Cancer Res* 68, 5591-5598 (2008).
2. Foltyn, M. et al. The physiological mTOR complex 1 inhibitor DDIT4 mediates therapy resistance in glioblastoma. *Br J Cancer* 120, 481-487 (2019).
3. Roos, G., Brattsand, G., Landberg, G., Marklund, U. & Gullberg, M. Expression of oncoprotein 18 in human leukemias and lymphomas. *Leukemia* 7, 1538-1546 (1993).
4. Chen, C.H. et al. Upregulation of MARCKS in kidney cancer and its potential as a therapeutic target. *Oncogene* 36, 3588-3598 (2017).
5. Ziemia, B.P., Burke, J.E., Masson, G., Williams, R.L. & Falke, J.J. Regulation of PI3K by PKC and MARCKS: Single-Molecule Analysis of a Reconstituted Signaling Pathway. *Biophys J* 110, 1811-1825 (2016).
6. Inozume, T. et al. Dendritic cells transduced with autoantigen FCRLA induce cytotoxic lymphocytes and vaccinate against murine B-cell lymphoma. *J Invest Dermatol* 127, 2818-2822 (2007).
7. Li, F.J. et al. FCRL2 expression predicts IGHV mutation status and clinical progression in chronic lymphocytic leukemia. *Blood* 112, 179-187 (2008).
8. Rodig, S.J. et al. The pre-B-cell receptor associated protein VpreB3 is a useful diagnostic marker for identifying c-MYC translocated lymphomas. *Haematologica* 95, 2056-2062 (2010).

5. The role of 17q gain in the refractory MCL sample, acquired in B4 clones, might be indeed interesting. The authors identified a list of 55 genes that were significant overexpressed in B4 versus all other samples (both from B patient and other analyzed patients). However, analyzing scRNA-Seq heatmap (Fig.3e) as well as Supplementary Table 2, it seems that there might be some differences within subpopulations of B4 samples (for example between B4a and B4b). Therefore, there might be some other differences between these two subpopulations that are not necessarily related to 17q gain. Furthermore, if I understood properly the data shown in Suppl. Table 2, only 22% of all cells from B4 samples showed overexpression of BIRC5, thus the link of this gene and 17q gain in non-responsive fraction does not seem that strong. In the line of this observation, the validation of BIRC5 expression levels in an additional series of non-responsive and responsive MCL samples are borderline and not very robust. It looks to me that the reported difference is due to the low expression (approx.. 3 FPKM) of one outlayer from resistant MCLs. Taken together, I believe that the authors should be more careful in drawing a general conclusion about the role of 17q gain and in particular of BIRC5 in the responsiveness to treatment.

We thank the reviewer for this excellent comment and have performed additional analyses. We identified significantly different CNVs between B4a and B4b and found that 17q gain was one of the regions that can distinguish these two subpopulations. Actually, 17q gain was more frequently observed in the B4b than the B4a subpopulation (53% vs. 35%, fisher test $P=7.04e-07$) (Fig. R4, left). BIRC5 (encodes Survivin), located at 17q, is highly expressed by the B4b subpopulation (Fig. 5a in revised manuscript), especially in the B4b cells at G2/M phase (Fig. R4, middle). The fraction of Survivin+ cells was present in 0.3% of the B4a population, 45.6% of the B4b population, and 36.7% of the B4b cells at G2/M phase, and Survivin expression was strongly associated with 17q gain (fisher test $P = 2.23e-10$, Fig. R4, right). Our observation is in line with the previous studies showing that survivin inhibits cell apoptosis and promotes cell proliferation at G2/M. Of note, as described above, expression of BIRC5 is highly regulated during cell cycle and is only expressed during G2/M phase. One would expect that given the sample collection and continuous but dynamic cell cycling stages, fractions of cells would be at different cell cycle stages including G0/G1, S and G2/M. Therefore, it is not surprising that not all cells with 17q gain express BIRC5 at a given time point of collection. We added this to Supplementary Figure 9.

Fig R4. 17q gain and BIRC5 overexpression. 17q gain (left) and BIRC5 expression (middle) in MCL cells of B4. The analysis was stratified by cell population (B4a and B4b) and cell cycle stage. The alluvial plot (right) demonstrates relationship between 17q gain and BIRC5 expression .

6. In relation to the above-mention studies of BIRC5, I found it difficult to understand why the Z138 cell line was not included in the cell line panel to test survivin inhibitor YM155. As authors mentioned previously, they found 17q gain only in this cell line, thus this should be the best model to check their hypothesis.

We thank the reviewer for this great comment. We performed an independent cell viability assay and included the Z138 cell line (together with 7 other MCL cell lines) and

found that Z138 is actually the most sensitive cell line to YM155 (Fig R5A-B), likely due to high expression of survivin in this cell line (Fig R5C). We have included the results in the revised manuscript (Fig. 5e).

Fig R5. Assessment of in vitro sensitivity to YM155 and surviving expression in MCL cell lines. (A-B) The *in vitro* efficacy of survivin inhibitor YM155 in 8 MCL cell lines. YM155-induced cell toxicity in MCL cell lines (red: ibrutinib-resistant; blue: ibrutinib-sensitive) in a dose (A)- and time (B)-dependent manner. (C) Western blot data to detect surviving expression in 8 MCL cell lines.

7. The single-cell techniques give a great opportunity to study clonal dynamics in the course of disease progression and theoretically, to detect even minor subpopulation that may arise at different time points. Unfortunately, I feel that the authors did not exploit sufficiently the potential of the data that they've generated. For instance, sometimes different subpopulation don't have to form a completely separated "cloud" in t-SNE plot, but may be distinguished by different level of expression of some genes. This kind of analysis would give a lot of potentially interesting information about heterogeneous landscape of MCL patients at different time point. Although I recognize that the authors have made some effort to track the clonal evolution, it was done in a superficial way and only in two patients. A more profound analysis of clonal evolution could be performed.

We thank the reviewer for this important comments. First, the clonal evolution analysis was limited by the sample size and the presence of tumor cells in sequential samples. Due to fast response to ibrutinib treatment, tumor cells in the PB samples from patients C and D were cleared quickly and very few tumor cells were detected by scRNA-seq. We therefore focused our clonal evolution analysis on patient B and V, which had good amounts of tumor cells for us to perform clonal evolution analysis.

Second, to address the reviewer's question regarding DEGs analysis beyond T-SNE plot, we applied SC3 (PMID: 28346451), an independent approach for unsupervised single-cell consensus clustering analysis (independent of tSNE DEGs analysis) and observed very similar results (Fig. R6, right). Cells of V3 were clearly separated from cells of V0, V1, V2, V5, which were clustered together with similar features.

Interestingly, the V0/1/2 tumor cells were likely eliminated by ibrutinib at the time point of V3 collection as no cells were detected in the sample V3 that exhibited similar expression features with cells of V0/1/2. Similarly, the vast majority of the V3 tumor cells might have been cleared from peripheral blood at the time point of V4 collection as only a small fraction of cells remained (Fig. R6, left, the V4 cells that clustered together with cells of V3). We added this to Fig. 3 in revised manuscript.

Fig. R6. t-SNE plots (left) and SC3 clustering (right) showing the cellular and transcriptomic characterization of the spleen compartment shift of tumor cells during treatment in patient V.

Reference:

1. Kiselev, V.Y. et al. SC3: consensus clustering of single-cell RNA-seq data. Nat Methods 14, 483-486 (2017).

8. *The discussion contains passages of overinterpretation, and the results should be interpreted and discussed within the limitations of the study design, that should be recognized and openly discussed.*

We thank the reviewer for this critique and have revised the language in the discussion section and also included comments on the limitation of this study.

Minor issues:

-MCLs are classified into conventional and leukemic non-nodal according to the WHO classification. Can the authors provide the SOX11 status and IGHV somatic hypermutation of the patients. A better description of clinico-biological features of the patients would be important.

IHC staining data showed positive SOX11 expression in patient B, D and E, while IHC staining data was not available for patient V and C. Due to lack of DNA sequencing data, we checked SOX11 expression in these patients using scRNA-seq. Non-responsive patient B showed the highest expression of SOX11 (Fig R7). Compared to patient B, responsive patients C, D and V showed lower SOX11 expression. We added the SOX11 status to Supplementary Table 1. As no samples were available for additional experiments (running out), we were not able to perform assay as suggest to analyze IGHV somatic hypermutations for these patients.

Fig R7. The proportion of cell expressing SOX11.

- Why heatmap 3e shows only around 18 genes? The authors mention that they determined 55 differential genes.

We identified 55 DEGs and selected the top most significant DEGs (\log_{10} p-value > 150, fold change > 0.6) to show in Fig. 3e (Fig.4e in revised manuscript). We have included the heatmap showing the full list of 55 genes, please see below (Fig. R8).

Fig R8. The heatmap of differential expressed genes.

- The color code is not helping in understanding several figures, e.g. Suppl. Fig1a, headings in the heatmaps in Suppl. Fig2. Please make sure the color codes clarify rather than confuse the reader.

We thank the reviewer for pointing this out and have revised the figures accordingly (Supplementary Fig. 2a and 5c in revised manuscript).

Reviewer #2 (Remarks to the Author): Expert in lymphoma

MCL is an aggressive B-cell lymphoma with poor prognosis due to emergence of drug resistant populations and lymphoma progression. Ibrutinib, a new FDA-approved bruton's tyrosine kinase (BTK) inhibitor, was shown to have high response rates in MCL patients. However, as the use of this drug continues to grow in MCL and other B-cell lymphomas, emergence of drug resistance and fatal progression are of increasing clinical concern. Remarkably, once MCL patients relapse after or on ibrutinib treatment there is rapid disease progression and patients die within 12 months. Thus, there is an urgent need to define mechanisms of ibrutinib resistance (IR) and to identify novel targets to bring forward novel treatment options with real curative potential for this fatal complication. Given that the mechanisms driving IR are poorly understood and no recurrent driver mutations have been identified in MCL, Zhang et al modeled IR evolution mechanism by implementing single-cell RNA sequencing in IR MCL lines and primary samples. They characterized MCL molecular heterogeneity and immune cellular diversity that drive drug resistance evolution and addressed an important clinical topic.

First of all, by using ibrutinib responders (n=3), non-responders (2) and control normal B-lymphocytes (2), scRNA-seq and WES, Zhang et al characterized molecular and cellular heterogeneity by patient, by response and cell type, and revealed that multiple cancer hallmark pathways (Myc, oxphos and mTOR) and acquisition of 17q were associated with IR evolution. These findings were further validated at genomic and cellular levels in extended primary patient samples and PDX models. Functionally, they showed that BIRC5/survivin, amplicon at 17q, upregulated and regulated cell survival and growth ex vivo and in vivo in resistant MCL tumor cells. Second, they explored the TME immunity in IR MCL samples and discovered significant different dynamics of CD8 T lymphocytes during ibrutinib treatment in ibrutinib responsive and IR patients and these changes were contributed to alteration of CD69 and CXCR4 expression. Overall the data show an interesting mechanism by which intratumor heterogeneity and TME immune cell dysregulation contribute to drug resistance evolution. Finally, the functional significance ex vivo experiments was validated by in vivo PDX MCL xenografts and primary samples. The paper is generally well presented with strong clinical correlation and bioinformatics analysis. Overall, the data shown are robust, convincing and the experiments well carried out.

However, to address some minor points will strength the manuscript. In discussion, a short explanation should be added to emphasize that, in addition to intratumor heterogeneity, tumor cell plasticity (transcriptome and kinome reprogramming) also play a critical role in IR evolution. Also, discussion on potential treatment options such as epigenetic modulation to prevent the onset of drug resistance.

We thank the reviewer for the positive feedback and the comments. We agree with the reviewer that tumor cell plasticity may also play a critical role in IR evolution, and epigenetic modulation could be a promising strategy to prevent/overcome drug resistance in MCL. We have revised our discussion accordingly.

Reviewer #3 (Remarks to the Author): **Expert in single cell sequencing**

Summary: In this manuscript, the authors performed longitudinal scRNA-seq analysis on PBMCs of mantle cell lymphoma (MCL) patients. They identified that tumor B cells in an ibrutinib non-responsive (NR) patient have unique cancer hallmarks distinct from that in responsive (R) patients, which was validated using a PDX mouse model. Furthermore, NR tumor B cells have 17q gain which induces the upregulation of survivin in NR patients compared to responsive patients. The authors also demonstrated that targeting survivin with YM155 is an effective therapeutic approach for MCL. Finally, they characterized differences in the tumor microenvironment between R and NR MCL patients.

The present manuscript is potentially important because it provides a potential mechanism of ibrutinib resistance in MCL. However, their claims are not convincingly supported by data and cannot be generalized since only one NR patient was considered.

Major points:

1. Figure 1c and 6a: The normal cell clusters of MCL patients are separated from that of healthy controls. The authors argued this suggests TME reprogramming, but the possibility of batch effects between MCL patients and healthy controls cannot be excluded. The authors used unusual high number of highly variable genes (9305) for clustering and dimensionality reduction, which might be a cause of the strong batch effects between MCL patients and healthy controls. I'm wondering whether reducing the number of highly variable genes or using batch correction methods resolves this issue.

We thank the reviewer for this comment. We observed minimal batch effects in our dataset (Fig. R9). We performed batch effect correction using fastMNN (PMID:Haghverdi L et al., Nature Biotechnology, 2018), but observed no significant difference in the tumor/immune cell composition pre and post batch effect correction (Supplementary Fig. 3b and c in revised manuscript). In addition, we tried clustering analysis using different number of variable genes (n=5000, 2000, 1000 genes, respectively) and got consistent results, demonstrating that TME cell subpopulations are clustered closely by cell type (Fig. R9).

Fig R9. t-SNE plots for raw data, batch effect corrected data and raw data based on different number of variable genes.

2. Figure 1d: In Patient E (NR), the fraction of tumor B cells is negligible and not detected in E2, which is not consistent with the clinical data presented in Figure 1b and the argument stating that the fraction of tumor B cells in NR patients increases during treatment. Since the authors used the cryopreserved samples, it might introduce selective cell losses. Most findings were derived from one NR patient (Patient B), which is difficult to generalize and the main weak point in this manuscript.

We thank the reviewer for this thoughtful comment and apologize for not having detailed description of the patient E in our original manuscript. For Single-cell RNA-seq, we included the pre-ibrutinib PBMC sample collected from Patient E (because of no residual BM specimens available at that time point). From the baseline PBMC sample,

we only captured 9 MCL cells, which were too few for subsequent analysis of tumor-intrinsic resistance factors. We therefore only used the non-malignant cells from patient E to help understand the TME related factors.

The flow cytometry data on Patient E showing 40-50% MCL cells (CD19+ and CD20+, Supplementary Table 1) was generated from a bone marrow sample. Although IGH CCND1 FISH analysis showed negative results for CCND1 translocation, IHC staining of informative markers showed positive expression of cyclin D1 in the bone marrow biopsies from this patient (Fig. R1). Flow cytometry immunophenotyping detected a monotypic B-cell population co-expressing CD5, CD19, CD20, CD22, CD38, CD44, CD79b and surface kappa light chain. The neoplastic cells are negative for CD3, CD4, CD8, CD10, CD11C, CD23, CD30, CD43, CD200 and lambda light chain. The primary diagnosis of patient E was confirmed through pathology re-reviewed and the results were verified independently by two experienced lymphoma pathologists. We included this in our revised manuscript (Page 5, Line 105-107).

Patient E

Fig. R1 CCND1 IHC Image for patient E.

We certainly agree with the reviewer on the limitation of the sample size. Given the small sample size, we performed multi-platform validation of the key findings at genomic and cellular levels in larger patient cohorts and also in PDX models. We have reframed our manuscript along the lines of a case report for patient B, still keeping other patients though at this time, as suggested by the editor. In addition, we have included language in the Discussion section on the limitation of this study.

Discussion, Page 21, Line 480-483:

“...However, these analyses were limited due to a small sample size. We therefore performed multi-platform validation of key findings at genomic and cellular levels in larger patient cohorts and also in PDX models...”

3. Figure 2d-f: The two cell subclusters of B4-PDX tumor cells should be indicated in Figure 2d,f and overlaid in Figure 2e. The robustness of trajectory analysis should be validated by using another pseudotime inference method.

We thank the reviewer for this comment and have performed additional analysis. We applied Scanpy, another pseudotime inference method. The results of Monocle and Scanpy are consistent, demonstrating that tumor cells from the B4-PDX model represented the spectrum of cellular and molecular heterogeneity that was similar to the parental B4 tumor cell populations. We have included this result as supplementary Figure 6.

Fig R10. Trajectory inference using Scanpy on OXPPOS and MTOR gene sets

4. Figure 3a,b: The normal cells should be included as a control. It seems that a subset of tumor B cells have normal CNV profiles, indicating that they might be normal B cells in MCL patients. This issue should be carefully examined. In Figure 3a, the CNV profiles of B4a and B4b look similar. However, they are segregated by the CNV profiles in Figure 3b. Which features in the CNV profiles determine the separation?

We indeed included normal B cells from two healthy donors as control for transcriptomic profiling. As suggested, we redid the inferCNV analysis by using the normal B cells as control and obtained very similar CNV profiles (Fig. 4a in revised manuscript). The reviewer was right that some tumor cells indeed showed low genomic instability and very few CNVs were observed, such as chr10, chr17 inV3 and chr1 in D1/2, which posts great challenge in tumor cell identification solely based on inferred CNVs. We therefore applied an integrative approach to identify malignant cells, which included inferred CNVs, cluster distribution and MCL related oncogenes expression.

We compared CNV profiles between B4a and B4b at chromosomal arm level. B4a and B4b subpopulations showed similar CNVs profiles across most of the chromosomes except 17q, 20p, 22q, 8p, in particular, 20p and 8p, where we observed significant difference (Fig.R11B).

A

B

Fig R11. Copy number variation estimation. (A) Inferred copy number profile using inferCNV. (B) Significantly different chromosomal regions between B4a and B4b cluster.

5. Figure 46b: Do the six NR patients have 17q gain compared to 15 R patients?

In the independent MCL cohort of n=21 patients (6 NRs, and 15 Rs), 17q was detected in one of the 6 NR patients but not in any of the 15 R patients analyzed. We identified 17q gain in additional refractory MCLs and in an intrinsically ibrutinib-resistant MCL cell line Z138 (Fig. 4d in revised manuscript), but we did not detect 17q gain in any of the responsive tumors tested.

6. Figure 4e: It seems that there is no difference of YM155 in vitro efficacy between ibrutinib-resistant and sensitive MCL cell lines, suggesting that YM155 is not specific to ibrutinib-resistant tumor B cells. Why is Rec-1 not responsive to YM155? Is survivin lowly expressed in Rec-1? How about the ibrutinib resistant Z-138 in Figure 2d? Do the ibrutinib-resistant cell lines have 17q gain and survivin overexpression compared to the sensitive cell lines? Does the combinatorial treatment of ibrutinib and YM155 increase the survival rate of MCL patients by targeting both sensitive and resistant clones?

We thank the reviewer for this comment. We agree with the reviewer that no significant difference was observed in the in vitro efficacy of YM155 between ibrutinib resistant and sensitive MCL cell lines. However, it seems like that YM155-induced cell toxicity correlated with the protein expression levels of Survivin. For example, Rec-1, the most resistant cell line to YM155, showed the lowest level of survivin expression (Fig. R12A and B). In line with this, Z138 and Jeko-1, the most sensitive cell lines to YM155, showed the highest levels of survivin expression (Fig. R12C). We updated Fig. 6c in revised manuscript.

Analysis of the WES data of cell lines showed that only Z-138 among all these cell lines had 17q gain. We also assessed the combinational effects of ibrutinib and YM155 via cell viability assay in the same eight MCL cell lines and did not observe synergistic effect (Fig. R12D). Therefore, we chose not to include the data in our revised manuscript.

Fig R12. Assessment of in vitro sensitivity to YM155 and **Survivin** expression in MCL cell lines. The *in vitro* efficacy of survivin inhibitor YM155 in 8 MCL cell lines. YM155-induced cell toxicity in MCL cell lines (red: ibrutinib-resistant; blue: ibrutinib-sensitive) in a dose (A)- and time (B)-dependent manner. (C) Western blot data to detect surviving expression in 8 MCL cell lines. (D) Combinational effect of ibrutinib and YM155 via cell viability assay in 8 MCL cells.

7. Figure 5e: The authors argued that V3/V4 tumor cells are new subpopulations redistributed from spleen, which are distinct from V0/1/2. However, V5 tumor cells, which should be more similar to V3/4, are clustered together with V0/1/2, which is contradictory to clonal evolution.

Ibrutinib has been shown in CLL and MCL to induce malignant cell redistribution from the tissue compartment (spleen and lymph node) into the PB during the initial weeks of therapy, a process also called ibrutinib-induced lymphocytosis, which is distinct from the process of clonal evolution observed in tumor B. In patient V (R), gradual splenomegaly shrinkage and multiple ALC peaks were observed following ibrutinib treatment at days 2 (V2), 10 (V2.5) and 22 (V3) after treatment (Fig. 3c in revised manuscript) and documented by the PET/CT imaging (Fig. 3a). Ibrutinib-induced lymphocytosis may lead to redistribution of transcriptomically distinct and similar tumor cell populations into the peripheral blood. The V0/1/2 tumor cells were likely eliminated by ibrutinib at the time point of V3 collection as no cells were detected in the sample V3 that exhibited similar expression features with cells of V0/1/2. Similarly, the vast majority of the V3 tumor cells might have been cleared from peripheral blood at the time point of V4 collection as only a small fraction of cells remained (the V4 cells that clustered together with cells of V3) (Fig. 3g, left). However, unlike V3/4, V5 tumor cells were clustered closer to V0/1/2, showing similar transcriptomic profiles. The V5 tumor cells may possibly represent the ibrutinib-responsive tumor cells as the V0/1/2 tumor cells, but further investigation in larger cohort will be needed.

8. Figure 6e: Are there any other gene set signature correlated with the proportion of CD8+ T cells?

We thank the reviewer for this query. Yes. We detected additional gene set signatures correlated with the proportion of CD8+ T cells, these gene set signatures included COMPLEMENT, IL6_JAK_STAT3_SIGNALING and PANCREAS_BETA_CELLS pathway signature in Hallmark, etc. (Fig. R13).

Fig. R13. The pathways associated with CD8+ T cells.

Minor points:

9. What is the cell type of the skyblue colored cluster in the bottom of Figure 1c?

We thank the reviewer for pointing this out. Those cells were erythroid progenitor cells, which was assigned a skyblue color on the tSNE plot but a purple color in the key. We apologize for this error and have made correction in the revised figure.

10. Line 150 “Overall, 13 cancer hallmarks were significantly upregulated in the ibrutinib-resistant tumors”: The authors should report the statistical significance of this sentence.

We thank the reviewer for this comment and have revised the manuscript as suggested.

11. Line 157-158 “significantly upregulated BIRC3”: Report the statistical significance.

We thank the reviewer for pointing this out. The adjusted p -value = 3.48×10^{-11} . We have added the p value in the revised manuscript (Page 9 Line 189).

12. Figure 2e, 4c, S2f: The corresponding gene set signature score should be also overlaid on the Monocle2 plot for better visualization.

We thank the reviewer for this comment and have revised the figure panels as suggested (Fig. 2e, 5b and Fig. S5e in revised manuscript).

13. Line 243: “high surviving expression highly” ◇ “high surviving expression IS highly”

We thank the reviewer for pointing this out. The word “surviving” should be “Survivin”. We have made corrections in the revised manuscript (Page 16 Line 362).

14. Line 254: “B2M checkpoint” ◇ “G2M checkpoint”

We thank the reviewer for pointing this out and have made corrections in the revised manuscript (Page 16 Line 353).

15. Figure 4d: The cell-cycle stage fraction of B4-PDX cells should be also shown for minor and major subclusters defined in Figure 2d.

We thank the reviewer for this comment and have revised the figure as suggested.

16. Line 277: Define “B2M”.

B2M (Beta-2-Microglobulin) is a prognostic marker for MCL and serves as an indicator of MCL tumor burden in mouse PDX models. We added the definition in revised manuscript (Page 17 Line 389-391).

17. Line 345 “we observed a significant decrease”: Report the statistical significance.

We thank the reviewer for this comment and have revised the manuscript as suggested. (Fig. 6b in revised manuscript).

REVIEWER COMMENTS

Reviewer #1 (Remarks to the Author):

The authors have adequately answered my questions.

Reviewer #3 (Remarks to the Author):

The revised manuscript addressed most of my concerns except the following points:

1. Pseudotime analysis: The added pseudotime analysis using Scanpy was poorly described. Scanpy is a general computational pipeline for analyzing scRNA-seq data, not specifically designed for pseudotime analysis. Since the authors did not explain how they perform the pseudotime analysis using Scanpy in the revised manuscript, I could not evaluate the technical soundness of their approach. Furthermore, the authors argued that the results of Monocle2 and Scanpy are consistent, but I found that B4a and B4b are placed on the same branch in Figure S6, which is not consistent with the result of Monocle2. In Figure S6, BM, PBMC, Liver, and Spleen should be grouped into subtype a and b like Figure 2e. Quantifying the difference of the pseudotime among conditions might be helpful for better visualization.

2. Figure R12: Figure R12C is useful for interpreting the difference of YM155 in vitro efficacy among cell lines, and should be included in the revised manuscript. I thought that 17q gain and Survivin overexpression are a key molecular event underlying the drug resistance of ibrutinib in MCL. Since there exists no combinatorial effect of ibrutinib and YM155, 17q gain and Survivin overexpression cannot be a driver of ibrutinib drug resistance in MCL. This should be carefully discussed.

Minor points:

1. Figure 1C: The cell type of the skyblue colored cluster is still missing. The authors used a wrong color in the color legend.

Dear Reviewers,

We would like to express our sincerest gratitude to all of you for your insightful reviews and constructive comments on our manuscript (NCOMMS-20-06359-B). Your insights and queries have helped us to significantly strengthen our manuscript. The manuscript has been substantially revised based on your reviews. Point-by-point responses to your comments are listed below

Reviewers' comments:

Reviewer #3 (Remarks to the Author):

The revised manuscript addressed most of my concerns except the following points:

- 1. Pseudotime analysis: The added pseudotime analysis using Scanpy was poorly described. Scanpy is a general computational pipeline for analyzing scRNA-seq data, not specifically designed for pseudotime analysis. Since the authors did not explain how they perform the pseudotime analysis using Scanpy in the revised manuscript, I could not evaluate the technical soundness of their approach. Furthermore, the authors argued that the results of Monocle2 and Scanpy are consistent, but I found that B4a and B4b are placed on the same branch in Figure S6, which is not consistent with the result of Monocle2. In Figure S6, BM, PBMC, Liver, and Spleen should be grouped into subtype a and b like Figure 2e. Quantifying the difference of the pseudotime among conditions might be helpful for better visualization.*

Thanks for your kind comment. We used three algorithms specifically designed for pseudotime inference: TSCAN, Slingshot and SCORPIUS. We respectively run these three tools with default parameters to infer pseudotime of tumor cells based on the hallmark gene sets of MYC, MTOR, G2M and OXPPOS signaling pathway. We regenerated trajectory structure using Monocle and mapped the pseudotime to trajectory. Three tools show that the inferred pseudotime of BM, PBMC, liver and spleen from PDX data are consistent with B4 sample (Fig. R1). In particular, subtype a and b of BM, PBMC, liver and spleen show consistent pseudotime with B4a and B4b (Fig. R1). We have replaced Figure S6 and revised the figure legend accordingly. We also described how to run these three algorithms in Methods (Page 27 line 619-623):

In order to check the robustness of pseudotime inference, we used three algorithms TSCAN⁶⁰, Slingshot⁶¹ and SCORPIUS⁶² which are specifically designed for pseudotime inference. We respectively run these three algorithms

with default parameters based on the same hallmark gene sets (MYC, OXPPOS, mTORC1 and cell cycle).

Reference

Zicheng Ji and Hongkai Ji. 2016. "TSCAN: Pseudo-Time Reconstruction and Evaluation in Single-Cell Rna-Seq Analysis." *Nucleic Acids Research* 44(13). Oxford University Press: e117-e117.

Street Kelly, Davide Risso, Russell B Fletcher, Diya Das, John Ngai, Nir Yosef, Elizabeth Purdom and Sandrine Dudoit. 2018. "Slingshot: Cell Lineage and Pseudotime Inference for Single-Cell Transcriptomics." *BMC Genomics* 19 (1). BioMed Central: 477.

Robrecht Cannoodt, Wouter Salens, Dorine Sichien, Simon Tavernier, Sophie Janssens, Martin Guilliams, Bart Lambrecht, Katleen De Preter, Yan Saeys. 2016. "SCORPIUS improves trajectory inference and identifies novel modules in dendritic cell development." *BioRxiv*.

Fig R1. Pseudotime inference using TSCAN, Slingshot and SCORPIUS (columns) based on the hallmark gene sets of G2M, MTOR, MYC and OXPHOS signaling pathway (rows). Colors are scaled by pseudotime inferred from each algorithm.

2. Figure R12: Figure R12C is useful for interpreting the difference of YM155 in vitro efficacy among cell lines, and should be included in the revised manuscript. I thought that 17q gain and Survivin overexpression are a key molecular event underlying the drug resistance of ibrutinib in MCL. Since there exists no combinatorial effect of ibrutinib and YM155, 17q gain and Survivin overexpression cannot be a driver of ibrutinib drug resistance in MCL. This should be carefully discussed.

Thank you for the great comment. We have included the data in R12C in the revised manuscript as Supplemental Fig S10b and revised the figure legend in for Fig S10 accordingly. We also revised the statement at line 367-368 accordingly:

These results indicate a crucial role of Survivin in contributing to MCL progression and resistance.

Minor points:

1. Figure 1C: The cell type of the skyblue colored cluster is still missing. The authors used a wrong color in the color legend.

Thanks for the kind comment. We have revised the color legend in Figure 1C.

REVIEWERS' COMMENTS

Reviewer #3 (Remarks to the Author):

My concern has been addressed by the authors.